# Systemic Resistance in Chilli Pepper against Anthracnose (Caused by *Colletotrichum truncatum*) Induced by *Trichoderma harzianum*, *Trichoderma asperellum* and *Paenibacillus dendritiformis*

**DOI:** 10.3390/jof7040307

**Published:** 2021-04-16

**Authors:** Mukesh Yadav, Manish Kumar Dubey, Ram Sanmukh Upadhyay

**Affiliations:** 1Laboratory of Mycopathology and Microbial Technology, Centre of Advanced Study in Botany, Institute of Science, Banaras Hindu University, Varanasi 221005, Uttar Pradesh, India; mkmkdubey@gmail.com (M.K.D.); upadhyay_bhu@yahoo.co.uk (R.S.U.); 2Department of Biosciences, School of Basic & Applied Sciences, Galgotias University, Greater Noida 203201, Uttar Pradesh, India

**Keywords:** biocontrol agents, *Capsicum annuum*, *Colletotrichum truncatum*, plant defence, biopriming, reactive oxygen species

## Abstract

In the present study, *Paenibacillus dendritiformis*, *Trichoderma harzianum,* and *Trichoderma asperellum* were appraised as potential biocontrol agents that induce resistance in chilli (*Capsicum annuum*) against the devastating pathogen *Colletotrichum truncatum,* which causes anthracnose. Bright-field and scanning electron micrographs showed the hyphal degradation, lysis, and abnormal swelling in *C. truncatum* against *P. dendritiformis* in a dual plate assay. Under greenhouse conditions, chilli seeds pretreated with *P. dendritiformis, T. asperellum, T. harzianum,* and *T. asperellum* + *T. harzianum* by soil soak method inflicted an induced systemic resistance (ISR) in chilli against a *C. truncatum*-challenged condition. In chilli, the disease index percentage was significantly reduced in the *T. asperellum* + *T. harzianum*-treated seeds, followed by the *T. harzianum-*, *T. asperellum-,* and *P. dendritiformis*-treated seeds as compared to the untreated and challenged, respectively. Chilli seeds were primed with *T. asperellum* + *T. harzianum* (78.67%), which revealed maximum disease protection under the challenged condition, followed by *T. harzianum* (70%), *T. asperellum* (64%), and *P. dendritiformis* (56%) as compared to untreated and *C. truncatum*-challenged (6%) condition served as control. The seeds that were pretreated with biocontrol agents (BCAs) inflicted ISR against *C. truncatum* by enhancing the activity of defence-related enzymes (superoxide dismutase (SOD), peroxidase (POX), polyphenol oxidase (PPO), catalase (CAT), ascorbate peroxidase (APX), guaiacol peroxidase (GPX) and phenylalanine ammonia-lyase (PAL)), accumulating phenolic compounds, and increasing the relative chlorophyll content in chilli. Nitroblue tetrazolium (NBT) and 3,3′-Diaminobenzidine (DAB) stains were used to detect the accumulation of superoxide anion and hydrogen peroxide that appeared nearby the fungal infection sites. The accumulation of reactive oxygen species (O_2_^−^ and H_2_O_2_) in the pathogen-inoculated leaves was a maximum of 48 hpi, followed by *P. dendritiformis, T. asperellum, T. harzianum,* and *T. asperellum* + *T. harzianum* treated tissue upon *C. truncatum*-challenged condition as compared to the control. Overall, our results showed the potential of *T. harzianum, T. asperellum, and P. dendritiformis* as biocontrol agents that prevent infection by *C. truncatum* and inflict an induced systemic resistance in chilli by enhancing the biosynthesis of phenolic compounds, defence and antioxidative enzymes, and reducing the lesion development and reactive oxygen species accumulation. This is the first report of induced systemic resistance against anthracnose in chilli obtained by application of *T. harzianum*, *T. asperellum* and *P. dendritiformis*, through seed priming.

## 1. Introduction

Chilli (*Capsicum annuum* L.), commonly known as “red pepper”, is an important fruit that is considered an essential spice in India and various other countries. Chilli peppers are a rich source of fibres, protein, vitamin A, C, and E, capsaicin, capsidiol, capsaicinoids, capsochrome, potassium, and folic acid. Additionally, it is used in beverages and medicines [1,2]. Capsaicin possesses analgesic, anticancer, antimutagen, antibacterial, antioxidant, immunomodulatory properties, and it has the potential to prevent platelet aggregation. At present, India is the largest producer, consumer, and exporter of dried chilli in the world [3]. The major important chilli-growing Indian states are Andhra Pradesh, Karnataka, Maharashtra, Orissa, West Bengal, Rajasthan, Tamil Nadu, and Uttar Pradesh, covering more than 80% area of India [4]. According to a recent report, the production of dried and green chilli fruits in India has reached around 1.389 million t and 0.0679 million t, cultivated in an area of 0.797 million hectares [5,6]. Due to their medicinal effects and daily use, it needs to maintain the chilli fruit yield all over the world. However, in the last few years, the fruit yield suffered because of the attack of various diseases of biotic and abiotic natures. Fungal diseases, such as anthracnose, damping-off, Fusarium wilt, collar rot, dry root rot, and stem rot, are considered the major cause behind these losses. Among these diseases, anthracnose of chilli caused by *Colletotrichum truncatum* (Schwenitz) Andrus & W. D. Moore) alone caused 50% yield loss worldwide [7,8].

Anthracnose disease is caused by seed- or air-borne fungus that mainly infects the leaves and fruits. Moreover, it also affects seed germination, plant growth, and the development and fruit qualities [9]. In the genus *Colletotrichum*, there are around 32 species that were reported to be associated with anthracnose in *Capsicum* spp. [10]. According to a recent study, *C. truncatum* and *C. scovillei* are considered to be the most prevalent pathogens associated with anthracnose of chilli in Asia [10]. However, in India, a yield loss of 10–54.91% has been reported due to fruit rot (anthracnose) disease caused by the destructive pathogen, *C. truncatum,* alone [11]. Integrated disease management (IDM), such as cultural, biological, and chemical control methods, has been used to control anthracnose. The maximum reduction has been attained via the application of fungicides; however, this is untenable to agriculture and human health and provides resistance to pathogens [3]. The use of resistant varieties is inexpensive, effortless and the most effective means of controlling anthracnose. However, since there are no resistant cultivars that have evolved, it is very important to use biocontrol agents (BCAs) because they are cost-beneficial and environment-friendly, and they are a convenient approach for disease control [12].

Plant growth promoting bacteria (PGPB) and plant growth promoting fungi (PGPF) are efficient biocontrol agents to control the seed- and air-borne diseases of several crops [13,14,15]. These beneficial bacteria and fungi are involved in plant growth and development through root colonization, production of siderophores and hormones, and nutrient uptake mechanisms [16]. The multiple modes of action, such as antibiosis, synthesis of essential secondary metabolites, cell wall degrading enzymes, callose and lignin deposition in the cell wall, led to the decrease in disease severity and induction of systemic resistance in the plant via sending signals from root to the shoot system for long-lasting protection against invading pathogens as compared to a chemically treated plant [17]. *Trichoderma* spp. are soil inhabitants and have the potential to suppress the growth and development of soil-borne pathogens via multiple modes of action, such as mycoparasitism, production of cell wall degrading enzymes and antibiotics, competition for space and nutrients with rhizospheric competition, and the induction of systemic resistance in chilli, pea, chickpea, tomato, and other plants [18,19,20]. Jasmonic acid is the component of induced systemic resistance expressed under rhizospheric treatment of PGPB and PGPF that induces the synthesis of flavonoids, phenol, and the expression of pathogenesis-related (PR) proteins and reactive oxygen species (ROS) scavenging enzymes, such as superoxide dismutase (SOD) [20,21]. Jasmonic acid-mediated signalling plays a key role in host resistance in a pathogen-challenged condition via increases in the activity of defence-related enzymes, such as peroxidase (POX) and polyphenol oxidase (PPO), which are majorly involved in generating free radicals and lignin deposition [22]. The pretreatment of the chilli seeds with *Bacillus amyloliquefaciens* enhanced the activities of phenylalanine ammonia-lyase (PAL), which regulates the synthesis of phenolic compounds that lead to xylem cell wall lignification and the accumulation of phytoalexins that prevents the entry and proliferation of soil-borne and foliar pathogens [16,23]. The PR proteins such as chitinase (PR-3) and β-1-3 glucanase (PR-2) are proteins that induce resistance in plants to several pathogen attacks through the process of cell wall degradation, cytoplasmic leakage, and ultimately, cell death [24]. The biochemical defence response is indicated by increased activities of antioxidative enzymes, such as CAT, SOD, GPx and APx, which are involved in the scavenging of ROS (H_2_O_2_ and O_2_^−^) during a pathogen attack [25,26]. Moreover, the increase in biochemical defence enzymes (POX, PPO, and PAL) and antioxidative enzymes (SOD, CAT, APx, and GPx) activities, phenol, and the accumulation of ROS reveal the key role of these molecules in the host–pathogen interactions.

Several reports on resistance induction in chilli against anthracnose have been made [16,20,27]. However, there is still a pressing need for an innovative approach towards anthracnose disease management. Out of these approaches, one significant strategy is to induce systemic resistance in the host plant by employing its innate immunity. Therefore, the present work was carried out to verify if the induced systemic resistance mechanism is responsible for the capacity of effective plant growth-promoting microbes (*Trichoderma harzianum*, *Trichoderma asperellum,* and *Paenibacillus dendritiformis)* to control anthracnose in chilli caused by *C. truncatum* under greenhouse condition. This is the first report available in the literature on the induction of systemic resistance by *P. dendritiformis,* in combination with *T. harzianum* and *T. asperellum,* against anthracnose in chilli.

## 2. Materials and Methods

### 2.1. Isolation, Morphological and Cultural Characterization of the Pathogen

The fungal pathogen was isolated from the anthracnose-infected chilli fruit samples. The chilli fruit samples were collected from nurseries in Varanasi, Uttar Pradesh, India. The fruit lesions with approximately a 5 mm diameter were chopped into small slices by a sterilized blade. The surface was sterilized with 1% NaOCl for 1–2 min followed by treatment with 70% of ethyl alcohol for 30 s and washed thrice with sterile distilled water and left for air drying in laminar flow for 2 h. The slices were placed in sterilized Petri plates containing potato dextrose agar (PDA) medium (Himedia, Mumbai, India) mixed with streptomycin (0.03 g L^−1^) and chloramphenicol (0.05 g L^−1^). Finally, the plates were incubated at 27 °C for 7 days. After 7 days of incubation, the newly formed mycelia from the edge of plates were transferred and followed by subculturing onto fresh PDA medium at 27 °C [28]. The isolate was identified based on morphological (size and shape of conidia and acervuli formation) and cultural (colony colour, growth pattern, and diameter of mycelium) characterization [29]. Further, the isolate was maintained at 4 °C in the refrigerator.

### 2.2. Molecular Identification of the Pathogen

#### 2.2.1. Fungal Genomic DNA Extraction

The pathogen identity was also confirmed by subjecting it to molecular analysis. The extraction of fungal genomic DNA was performed using the CTAB method with modifications from a previous study [30]. For isolation of the fungal DNA, the isolate was grown for 7 days on PDA medium at 27 °C. The mycelia were scraped from the PDA with the sterile blade and placed in an autoclaved microcentrifuge tube (1.5 mL). The ~0.5 g of mycelium was frozen in liquid nitrogen. They were then ground into slurry form with the help of a micro pestle, transferred to 0.5 mL of extraction buffer [0.1 M Tris HCl (pH 8.0), 0.01 M EDTA (pH 8.0), 1.5 M NaCl], and mixed thoroughly. This was followed by incubation for 30 min at 65 °C. The DNA was extracted by the addition of equal volumes of phenol: chloroform: isoamyl alcohol (25:24:1). The tubes were vortexed and mixed thoroughly and gently for 10 min at 10,000 rpm. The supernatant was transferred to a new Eppendorf tube, and 0.3 mL of chilled isopropanol was added and spun for 10 min at 10,000 rpm to collect the pellet. The DNA pellet was washed with 70% ethyl alcohol and completely dried at RT. After that, the pellet was transferred to a new microcentrifuge tube containing the 0.05 mL of 1X Tris-EDTA buffer and stored at −20 °C for PCR amplification.

#### 2.2.2. PCR Amplification and Sequencing

Pathogen identification was based on the amplification of the ITS region using the sequences of primer pair ITS 1(5′TCCGTAGGTGAACCTGCGG’3)/ITS 4(5′TCCTCCGCTTATTGATATGC′3) [31]. The volume of 20 µL of polymerase chain reaction mixture containing 20−50 ng of DNA, 1.5 mM of MgCl_2_, 0.25 mM of each primer, 0.2 mM of d NTPs, 1× of the buffer, and 2 µL of Taq polymerase. PCR cycling amplification parameters were followed as the initial denaturation at 94 °C for 3 min, followed by denaturation at 94 °C for 30 s, 72 °C for 1 min with a final extension step for 10 min at 72 °C. The amplified PCR product was examined in 1% (*w*/*v*) agarose gel by electrophoresis and stained with ethidium bromide (0.5 mg/mL). The quality and quantity of the amplified PCR products were determined by the gel documentation system (Bio-Rad, Gurugram, India) and the Nanodrop Spectrophotometer. The sequencing of the purified PCR products was carried out at Chromus Biotech Pvt. Ltd. The sequence has been submitted to the GenBank database with NCBI accession no. MW541903.

### 2.3. Plant Growth Promoting Bacteria (PGPB) Isolates

The healthy and asymptomatic chilli roots were collected during a routine survey from agricultural fields in and around Varanasi, Uttar Pradesh, India (25.3176° N, 82.9739° E). The collected roots were cut approximately 0.5 mm × 0.5 mm in size and chopped into small slices by a sterilized blade. The surface was sterilized with 1% NaOCl for 1–2 min followed by treatment with 70% of ethyl alcohol for 30 s and washed thrice with sterile distilled water and left for air drying in laminar flow for 2 h. Afterward, 4–5 small slices of tissue were placed on Petri plates containing nutrient agar (NA) medium (Himedia, India) and incubated at 30 °C. The pure colony of isolated bacteria was transferred onto separate Petri plates from the mixed colony [32]. A total of 13 endophytic bacterial isolates were differentiated based on the differences in their colony morphology.

#### 2.3.1. Selection of Plant Growth Promoting Bacterium

All the 13 endophytic bacterial isolates were screened for their antagonistic activity against the phytopathogen (*C. truncatum*) by using the dual culture method [33]. Out of 13, only a single endophytic bacterium exhibited the strongest biocontrol activity against the phytopathogen and was selected and used for further experiments.

#### 2.3.2. Identification of the Selected Plant Growth Promoting Bacterium

The selected endophytic bacterium was identified as *Paenibacillus dendritiformis* based on 16S rDNA gene analysis [34] and submitted to the GenBank database with NCBI accession no. MW365553. The isolate was stored at −80 °C with glycerol (20%) stock for further use.

### 2.4. Collection of Plant Growth Promoting Fungi

The accessible microbial cultures *Trichoderma harzianum* BHU BOT RYRL4 (NCBI GenBank accession no-KR 856210) and *Trichoderma asperellum* BHU BOT RYRL1 (NCBI GenBank accession no-KR 856207) were obtained from the RY Roy Laboratory of Mycopathology, Department of Botany, Institute of Science, Banaras Hindu University, Varanasi, India. Then, the seven-day-old cultures of *Trichoderma* were maintained in PDA slant medium and stored at 4 °C in the refrigerator for six months and revived subsequently for use.

### 2.5. Preparation of Bacterial and Fungal Inocula

The isolated *P. dendritiformis* cell inoculum was prepared from the 24-h-old culture at 30 °C. The fresh culture of the bacterium was immersed in 10 mL of deionized H_2_O, scraped using a sterile inoculating loop, and maintained to a cell suspension of 1 × 10^8^ CFU/mL (OD_600_ = 0.8). In contrast, the conidial suspension of the *C. truncatum* was prepared using the method described previously [8]. The *C. truncatum* was grown in a Petri plate containing PDA medium for 21 days at 27 °C. Thereafter, the Petri plate was immersed in 10 mL of deionized water, and the conidia were scraped using an autoclaved glass spreader. The conidial inoculum was filtered using two layers of sterile muslin cloth and further, the filtered suspension was diluted with deionized water to adjust the final concentration to 1 × 10^6^ conidia/mL by counting with a hemocytometer. Likewise, the preparation of a conidial suspension of *T. harzianum* and *T. asperellum* was also done and adjusted to a final concentration of 2 × 10^7^ conidia/mL.

### 2.6. Plant Material and Growth

The susceptible seeds (*Capsicum annuum* cv. Surajmukhi) were obtained from the IIVR, Varanasi, India. The seeds were surface sterilized with 1% NaOCl for 1–2 min followed by treatment with 70% of ethyl alcohol for 30 s and washed thrice with sterile distilled water and left for air drying in laminar flow for 2 h. The surface-sterilized seeds were grown in mixed autoclaved soil (clay/vermicompost, 3:1, *v*/*v*) in a glasshouse condition of 14 h light and 10 h dark cycles at 27 ± 1 °C. After attaining a height of 15–20 cm (Up to 10 true leaves per shoot) and at the fruiting stage (>90% fruits have typical fully ripe colour), these plants were used for testing the pathogenicity of *C. truncatum* and also for further experiments.

### 2.7. Pathogenicity Test on Leaves and Fruits

A pathogenicity test was performed on 15–20 cm-long chilli plants (Up to 10 true leaves per shoot) by spraying the leaves with a conidial suspension of *C. truncatum* (pathogen) [35]. The optimum inoculum concentration was maintained at 1 × 10^6^ conidia mL^−1^ and sprayed on the leaf area. These plants were observed for 2–3 days for disease development. Similarly, the sterile distilled water sprayed on leaves serve as the control.

Further, the pathogenicity of *C. truncatum* was tested on ripe fruits (>90% fruits have typical fully ripe colour) of chilli (140-day-old plant) by using a pin-prick inoculation method [36] under greenhouse conditions at the botanical garden, Department of Botany, Banaras Hindu University, Varanasi, Uttar Pradesh, India. One set of fruits was inoculated with 10 μL of conidial suspension with the help of a sterile syringe, and the other set of fruits was sprayed with sterile distilled water, which served as a control. Both sets of fruits were regularly monitored for 7 days for disease development. The experiment was repeated thrice, and each set of treatments contained six pots.

#### Measurement of Disease Index (DI)

The disease index was measured after 3 and 6 days post-inoculation of the conidial suspension of *C. truncatum* on the chilli fruit, according to the method from a previous study [37]. The disease index was measured on a scale ranging from 0 to 3. The 0–3 scale of the disease index was categorized as follows:0 = No visible disease symptom on the fruit.1 = Slight infection, with small spots (≤1 mm).2 = Moderate infection, with medium spots (1–2 mm).3 = Severe infection, with large spots (>2 mm).

The disease index (DI) was calculated using the following formula [35]:$$\mathrm{Disease}\;\mathrm{Index}\;\left(\%\right)=\frac{(\sum_{}^{}\mathrm{scale}\times \mathrm{number}\;\mathrm{of}\;\mathrm{spots}\;\mathrm{per}\;\mathrm{fruit})}{\mathrm{Highest}\;\mathrm{scale}\times \mathrm{total}\;\mathrm{number}\;\mathrm{of}\;\mathrm{spots}\;\mathrm{per}\;\mathrm{fruit}}\times 100$$

Ten fruits per replication were maintained for each treatment, and the experiment was performed thrice. Fruits inoculated with sterile water served as control.

### 2.8. In Vitro Antagonism of P. dendritiformis, T. harzianum and T. asperellum against C. truncatum

In vitro antibiosis assay was carried out according to the method provided by Upadhyay and Rai [33]. The inhibition of *C. truncatum* growth by *P. dendritiformis, T. harzianum,* and *T. asperellum* was carried out on the PDA media using the dual culture technique. *T. harzianum, T. asperellum,* and *P. dendritiformis* mycelial plugs with a diameter of 5 mm were placed on the front side of the plate containing a *C. truncatum* mycelium plug with a diameter of 5 mm that was 2 cm away from the periphery and kept at 27 °C for 7 days. The inoculated *C. truncatum* (5 mm diameter) and sterile agar plugs on the opposite side of the plate served as control. The characteristics of the inhibition of *C. truncatum* growth in both the test and control were observed after 2 days of inoculation. The data were obtained for the percent inhibition of radial growth (PIRG) by using the formula (100 × (C − T)/C) where C = radial growth of the *C. truncatum* in control and T = radial *C. truncatum* growth in dual culture with the antagonist and the width of the zone of inhibition (ZI) [38]. The experiment was performed thrice in triplicates.

### 2.9. Scanning Electron Microscopy to Study the Interaction between P. dendritiformis and C. truncatum

The post-interaction between *P. dendritiformis* and *C. truncatum* based on the antibiosis assay was observed by a scanning electron microscope. The hyphae from the zone of interaction were taken out from a dual culture plate and transferred onto glass slides. The samples were fixed in 1.5% glutaraldehyde and kept for 12 h at 4 °C in a dry Petri dish. Subsequently, the samples were dehydrated with a graded series of ethanol (10%, 30%, 50%, 70%, 90%, and 100%) and dried in a desiccator [39]. Finally, the dried specimens were tape-affixed and subsequently coated with gold [40]. The specimens were observed at 20 kV in ZEISS, model number EVO-18 (Germany).

### 2.10. Experimental Design for Chilli Seed Treatments

The experiments were performed in the pot under greenhouse conditions (80% RH with a 14 h light and 10 h dark cycle at 27 ± 1 °C) at the botanical garden, Department of Botany, Institute of Sciences, BHU, Varanasi, India. The experimental research was categorized into untreated (control) plants and biocontrol agents-treated plants challenged with the pathogen. The experiment was comprised of 6 treatments, i.e., *P. dendritiformis*-treated seeds, *T. asperellum*-treated seeds, *T. harzianum*-treated seeds, and *T. asperellum + T. harzianum*-treated seeds, upon pathogen-challenged and pathogen-inoculated samples, whereas untreated and unchallenged seeds served as control. All the experiments were performed thrice in triplicates, and for each treatment, 3 pots containing 3 seedlings were maintained.

#### Seed Priming with *P. dendritiformis*, *T. asperellum*, *T. harzianum*, *T. asperellum* + *T. harzianum*

The inoculum suspension of *P. dendritiformis* (1 × 10^8^ CFU/mL), *T. harzianum,* and *T. asperellum* (2 × 10^7^ conidia/mL) were spun for 15 min at 10,000 rpm. The pellets were suspended in 100 mL of sterile deionized water containing 1.5 g carboxymethyl cellulose (CMC) [41]. The susceptible variety of chilli seeds (cv. Surajmukhi) were surface sterilized with 1% NaOCl for 1–2 min followed by treatment with 70% of ethyl alcohol for 30 s and washed thrice with sterile distilled water and left for air drying in laminar flow for 2 h. The sterilized seeds were soaked in an inoculum suspension of *P. dendritiformis*, *T. harzianum*, *T. asperellum,* and *T. asperellum + T. harzianum* on a shaker for 12 h at 150 rpm [42]. The soaked seeds were filtered and dried on sterile blotting paper in laminar airflow. The CMC-soaked seeds devoid of inoculum suspension of biocontrol agents were served as control.

### 2.11. Treatment of Chilli Plants with Biocontrol Agents under Greenhouse Condition

The bioprimed seeds were grown in pots (15 cm × 10 cm) containing 1.5 kg mixed autoclaved soil (clay/vermicompost, 3:1, *v*/*v*) (rhizospheric application of bioagents). Afterward, soil drenching and foliar spraying (phyllospheric application of bioagents) were carried out to treat the seedlings with *P. dendritiformis*, *T. asperellum*, *T. harzianum,* and *T. asperellum + T. harzianum* five times in the entire life cycle of the plant. After the treatments with bioagents, at the fruiting stage, the fruits were sprayed with the conidial suspension of *C. truncatum* and covered using autoclaved plastic bags for 96 h to retain moisture. The untreated and unchallenged chilli fruits were sprayed with deionized water, served as control, and covered for 96 h to retain humidity.

### 2.12. Assessment of P. dendritiformis, T. asperellum, T. harzianum, and T. asperellum + T. harzianum on Fruit Anthracnose Protection against C. truncatum under Greenhouse Conditions

The seeds were treated with the inoculum suspension of *P. dendritiformis, T. asperellum, T. harzianum,* and *T. asperellum + T. harzianum* under pathogen-challenged condition and pathogen sprayed, whereas the untreated and unchallenged served as control. The treatment of biocontrol agents was performed through soil drenching and repeated up to five times before the challenge of the pathogen on fruits. At the end of 120 days, old plants containing green chilli fruits were surface sterilized and sprayed with the conidial suspension of *C. truncatum*. For each treatment of six pots, each pot containing two plants was set up, and the experiment was performed thrice in triplicates. The fruits were examined for the development of anthracnose symptoms for 10 days after pathogen inoculation (DPI). The percent of fruit rot disease protection was calculated using the mentioned formula [16].
$$\mathrm{Disease}\;\mathrm{Protection}\;\left(\%\right)=\frac{\mathrm{Number}\;\mathrm{of}\;\mathrm{infected}\;\mathrm{fruits}}{\mathrm{Sum}\;\mathrm{of}\;\mathrm{fruits}\;\mathrm{per}\;\mathrm{plant}}\times 100$$

### 2.13. Chlorophyll Estimation

The chilli seeds were treated with bioagents under the *C. truncatum*-challenged condition to assess the anthracnose development by observing the changes in chlorophyll content (chlorophyll a, b and total chlorophyll). The chlorophyll content was estimated using Arnon’s method [43]. Here, five hundred milligrams of green chilli fruit was ground in 5 mL of 80% acetone, and the extract was spun for 10 min at 10,000 rpm. The aqueous layer was transferred to another tube, and the absorbance was recorded spectrophotometrically at 645 and 663 nm for chlorophyll a, b and total chlorophyll, respectively. The chlorophyll content was calculated using the mentioned formula [44].
Chlorophyll a (mg/g) = 12.7 (OD663) − 2.69 (OD645) × V/1000 × W
Chlorophyll b (mg/g) = 22.9 (OD645) − 4.68 (OD663) × V/1000 × W
Total chlorophyll a (mg/g) = 20.2 (OD645) + 8.02 (OD663) × V/1000 × W
where OD = optical density, W = weight of sample, and V = volume of 80% acetone (mL).

### 2.14. Biochemical Studies

#### 2.14.1. Fruit Harvest and Analysis

One hundred twenty days after sowing (DAS), the chilli fruits were challenged with *C. truncatum* (1 × 10^6^ conidia mL^−1^) under greenhouse conditions at 27 ± 1 °C. The 120-day-old chilli fruits treated with bioagents under the pathogen-challenged condition, the untreated and unchallenged chilli fruits served as negative control, and the untreated, pathogen-challenged chilli fruits served as positive control were all harvested at different meantimes such as 0, 12, 24, 48, 72, and 96 hpi. The harvested chilli fruits were stored at −80 °C for further experiments. All the experiments were performed thrice in triplicates for each treatment.

#### 2.14.2. Defence Enzyme Assay

##### Assessment of Peroxidase (POX) Activity

The POX activity was performed using the method from a previous study [45]. In this assay, 500 mg of chilli fruit was ground in a chilled mortar and pestle containing 3 mL of 100 mM sodium phosphate buffer (pH 7.0) at 4 °C. The extracted solutions were spun for 15 min at 16,000 rpm. The aqueous solution was used as an enzyme extract. The 2.5 mL reaction mixture was prepared using 1.5 mL of 50 mM pyrogallol, 0.5 mL of 1% hydrogen peroxide, and 0.5 mL crude enzyme extract. The changes in optical density were recorded spectrophotometrically at 470 nm for 1 min. The peroxidase activity was expressed as min^−1^ mg^−1^ protein (extinction coefficient = 26.6 mM^−1^ cm^−1^). The experiment was performed thrice in triplicates for each treatment.

##### Assessment of Phenylalanine Ammonia-Lyase (PAL) Activity

The PAL activity was performed using the method from a previous study [46]. In this method, 500 mg of chilli fruit was ground with 3 mL of 100 mM borate buffer (pH 8.7) in a chilled mortar and pestle. The homogenized extract was centrifuged at 10,000 rpm for 25 min at 4 °C, and the upper aqueous layer served as enzyme extract. The reaction mixture consisted of 0.5 mL of 12 mM phenylalanine ammonia-lyase, 1.3 mL of 100 mM borate buffer (pH 8.7), and 0.2 mL enzyme extract. The reaction mixture was incubated at 40 °C for 1 h in a water bath. The increase in optical density was recorded at 290 nm, and the PAL activity was exhibited as the synthesis of µmol trans-cinnamic acid min^−1^ mg^−1^ protein. The experiment was performed thrice in triplicates for each treatment.

##### Assessment of Polyphenol Oxidase (PPO) Activity

The PPO activity was assessed by the following protocol from a previous study [47]. Here, 500 mg of chilli fruit was crushed in 2 mL of 0.1 M sodium phosphate buffer (pH 6.5). The crude extract was spun for 15 min at 16,000 rpm, and the supernatant was served as enzyme extract. The reaction mixture consisted of 1.5 mL of 100 mM sodium phosphate buffer (pH 6.5), 200 µL 10 mM catechol, and 200 µL of enzyme extract. The increase in optical density was measured at 495 nm for 1 min. The PPO activity was exhibited as min^−1^ mg^−1^ protein. The experiment was performed thrice in triplicates for each treatment.

#### 2.14.3. Antioxidant Enzyme Activities

##### Assessment of Superoxide Dismutase (SOD) Activity

The assessment of SOD activity was carried out according to the method from a previous study [48]. In this method, 0.5 g of chilli fruit was crushed in 5 mL of 100 mM sodium phosphate buffer (pH 7.5) contained 0.5 mM EDTA. The homogenized chilli tissue was spun for 13,000 rpm for 20 min. The supernatant obtained served as crude enzyme extract. In a test tube, 3 mL of the reaction mixture was prepared with a 0.05 M sodium phosphate buffer (pH 7.8), 75 µM nitro blue tetrazolium chloride, 100 mM EDTA, 13 mM methionine, 60 µM riboflavin, and 0.1 mL enzyme extract. The tubes were exposed to a 400 W bulb (4 × 100 W bulbs) for 15 min at 25 °C. The optical density was recorded at 560 nm, and the activity of SOD was exhibited as the unit’s g^−1^ fw. The experiment was performed thrice in triplicates for each treatment.

##### Assessment of Catalase (CAT) Activity

The activity of catalase was determined using the method from a previous study [49]. For crude enzyme extraction, 500 mg of chilli fruit was crushed in 3 mL of 0.05 M Tris HCl buffer (pH 8.0) comprising of 0.5% *v*/*v* Triton X100, 2% *w*/*v* polyvinylpyrrolidone, and 0.5 mM EDTA. The extract was spun for 15 min at 13,000 rpm, and the resulting aqueous solution served as the enzyme extract. The chemical reaction was initiated by the addition of 300 µL of crude extract in 1.5 mL of 0.1 M sodium phosphate buffer (pH 7.0) and 1.2 mL of 150 mM hydrogen peroxide. The tubes were placed in dark conditions for 60 s, and the absorbance was measured at 240 nm. The change in optical density as the H_2_O_2_ was converted into molecular oxygen. The activity catalase was exhibited as nmol min^−1^ mg^−1^ protein. The experiment was performed thrice in triplicates for each treatment.

##### Assessment of Ascorbate Peroxidase (APx) Activity

The APx activity was assessed by using the method from a previous study [50]. Here, 500 mg of chilli fruit was macerated in 0.1 M sodium phosphate buffer (pH 7.0) in a chilled mortar and pestle. The macerated chilli tissue was spun for 15 min at 13,000 rpm, and the resulting aqueous solution was served as an enzyme source. The reaction mixture comprised of 1.5 mL of 0.1 M sodium phosphate buffer (pH 7.0), 300 µL of 0.005 M ascorbic acid, 600 µL of 0.5 mM hydrogen peroxide, and 600 µL of crude extract. The decrease in optical density was measured spectrophotometrically at 290 nm after 60 s and the addition of the enzyme extract. The activity of APX was exhibited as nmol ascorbate oxidized min^−1^ mg^−1^ protein (extinction coefficient = 2.8 mM^−1^ cm^−1^). The experiment was performed thrice in triplicates for each treatment.

##### Assessment of Guaiacol Peroxidase (GPx) Activity

The GPx activity was performed according to the method from a previous study [51]. For enzyme extraction, 0.5 g of chilli fruit was ground in 3 mL of 0.1 M sodium phosphate buffer (pH 7.0). The extract was spun for 15 min at 13,000 rpm, and the obtained supernatant was used as the enzyme extract. In a test tube, the chemical reaction began after the addition of 100 µL of crude enzyme extract in mixtures of 3.0 mL of 0.1 M sodium phosphate buffer (pH 7.0), 30 µL of 12.3 mM hydrogen peroxide, and 50 µL of 20 mM guaiacol solution. The decrease in optical density was measured per unit minute at 436 nm. The guaiacol peroxidase activity was exhibited as µmol min^−1^ mg^−1^ protein. The oxidation of 1 µmol guaiacol per minute indicates one unit of GPx activity (extinction coefficient = 25 mM^−1^ cm^−1^). The experiment was performed thrice in triplicates for each treatment.

#### 2.14.4. Estimation of Total Phenolic Compound

The total phenolic compound was estimated according to the method from a previous study [52]. In this method, 0.5 g of chilli fruit was crushed in 10 mL of 80% of methanol and incubated for 15 min at 70 °C in a water bath. The homogenate was spun for 20 min at 13,000 rpm. The aqueous solution was obtained and used as a phenolic extract. The reaction mixture comprised of 1 mL phenolic extract, 5 mL of DDW, and 0.25 mL of 1 N Folin–Ciocalteu reagent and kept for half an hour at 25 °C. The change in absorbance was recorded at 725 nm using catechol as a standard. The phenolic content was exhibited as µg of catechol produced g^−1^ of fw. The experiment was performed thrice in triplicates for each treatment.

#### 2.14.5. Histochemical Analysis

##### Detection of Hydrogen Peroxide Accumulation

The accumulation of hydrogen peroxide on the leaf surface was detected using the DAB staining method from a previous study [53]. The chilli leaves treated with biocontrol agents under *C. truncatum*-challenged condition and the untreated and unchallenged (control) samples were collected at 48 h after pathogen inoculation, and then the samples were used for observation of histochemical accumulation of ROS-like H_2_O_2_. The treated and untreated leaves were detached from the plant and dipped in 1 mg mL^−1^ of the DAB solution and maintained at a pH 3.8 using 0.1 N HCl. The leaves were kept for 8 h in dark condition. After incubation, the leaves were placed on filter paper and then boiled in 96% ethanol till the chlorophyll pigment was completely removed. The histochemical appearance of the reddish-brown colour on the leaf surface indicates the hydrogen peroxide deposition, which was observed and photographed under an Olympus binocular microscope. The experiment was performed thrice in triplicates for each treatment.

##### Nitroblue Tetrazolium (NBT) Staining to Detect O_2_^−^ Deposition

The superoxide anion (O_2_^−^) deposition was detected using the NBT staining method from a previous study [54]. The chilli leaves treated with biocontrol agents under pathogen-challenged conditions and the untreated and unchallenged samples were collected 48 h after pathogen inoculation, and then the samples were used for the observation of O_2_^−^ accumulation. The leaf samples were immersed in an amber-coloured bottle containing 100 mg of NBT in 0.05 M sodium phosphate buffer (pH 7.5) and made a final volume of 50 mL to obtain a 0.2% NBT solution. The untreated and unchallenged (control) leaves and the treated leaves were kept for 8 h in dark condition. After 8 h of incubation, the leaves were placed on filter paper and then boiled in 96% ethanol till the complete removal of chlorophyll. The dark blue colour on the leaf tissue revealed the superoxide anion (O_2_^−^) accumulation and was observed under an Olympus binocular microscope. The experiment was performed thrice in triplicates for each treatment.

### 2.15. Statistical Analysis

All the presented data were examined applying the analytical software SPSS ver. 16.0. The experiments were done in triplicates of each treatment, and the values were examined using one-way ANOVA. The significant mean separation of each data was assessed by using Duncan’s multiple range test at *p* values ≤ 0.05. Principal component analysis (PCA) was performed with PAST for Windows 10 software.

## 3. Results

### 3.1. Morphological, Cultural and Molecular Characterization of the Pathogen

The isolated pathogen from the chilli fruit associated with anthracnose disease was identified based on its structural and cultural characteristics. The isolate has an upper dark grey and reverse dark brown mycelia, cottony growth, and the mycelium grew at 4 mm/day in PDA medium. The conidia were hyaline, aseptate, sickle-shaped, tapering at both ends, and measured ~23.8 µm × 3.6 µm. The concentric ring of black acervuli, asexual fruiting body in which the conidia were situated on a conidiophore and sterile septate setae were also observed. Based on the morphological, cultural, and molecular characterization the isolated fungus was identified as *C. truncatum* (Figure 1).

### 3.2. Pathogenicity Test

The chilli leaves and fruits sprayed with the conidial suspension of *C. truncatum* produced anthracnose disease symptoms, such as dark black, circular, and sunken lesions with the concentric ring of acervuli, after seven days of incubation. In vivo pathogenicity test using chilli cv. “Surajmukhi”, necrotic lesions appeared in all treated fruits and leaves. In contrast, the control fruits and leaves sprayed with sterile distilled water showed no anthracnose symptoms or lesions. The isolates that were re-isolated from the fruits and leaves infected with *C. truncatum* showed similar morphology and cultural characteristics as that of the original isolate, thereby confirmed Koch’s postulates (Figure 2).

### 3.3. Disease Index

Anthracnose lesions in chilli fruits were measured after 3 and 6 days post-inoculation of the *C. truncatum* conidial suspension. The in vivo length of the developed lesion was calculated in pathogen-inoculated and uninoculated chilli fruits using the formula of disease index (DI) (Figure 3).

### 3.4. Dual Culture Assay

In this study, *P. dendritiformis, T. harzianum,* and *T. asperellum* were assessed for their antagonistic activity against *C. truncatum*. Among them, *T. harzianum, T. asperellum,* and *P. dendritiformis* were screened based on their efficient biocontrol activity. The PIRG of *C. truncatum* varied from 71.56% to 75.46% (Table 1). In vitro antagonism of *P. dendritiformis, T. harzianum,* and *T. asperellum* inhibits the hyphal growth of *C. truncatum*, a destructive pathogen that causes qualitative and quantitative yield losses in chilli. Among the screened antagonists, *T. harzianum* showed a maximum of 75.46% radial growth inhibition against *C. truncatum*, followed by *T. asperellum* (73.09%) and *P. dendritiformis* (71.56%). The antagonistic activity of PGPB and PGPFs against *C. truncatum* was performed by employing dual culture plates (Figure 4 and Figure 5).

### 3.5. Scanning Electron Microscope Observation

The hyphae from the zone of interaction of the 5-day-old antagonistic plate was observed in a scanning electron microscope. The *P. dendritiformis* caused the hyphal lysis, degradation, abnormal swelling, disintegration, and leakage of cytoplasmic contents in *C. truncatum* as compared to the untreated (control) sample (Figure 6).

### 3.6. Assessment of P. dendritiformis, T. harzianum, T. asperellum, and T. harzianum + T. asperellum on Fruit Anthracnose Protection against C. truncatum under Greenhouse Conditions

The chilli seeds primed with *P. dendritiformis*, *T. harzianum, T. asperellum,* and *T. harzianum + T. asperellum* under greenhouse conditions reduced the development of anthracnose lesions on fruits. Among the different combinations used, the seeds treated with bioagents, *T. harzianum* + *T. asperellum,* showed a maximum reduction of 78.67% in anthracnose disease development followed by *T. harzianum, T. asperellum,* and *P. dendritiformis* under the challenged condition as compared to the untreated and challenged (control), wherein 94% of chilli fruits infected with *C. truncatum* (Figure 7 and Figure 8).

### 3.7. Effect of P. dendritiformis, T. harzianum, T. asperellum and T. harzianum + T. asperellum on Chlorophyll Content

The chilli fruits treated with *P. dendritiformis, T. harzianum,* and *T. asperellum* enhanced the total chlorophyll, chlorophyll a, and chlorophyll b content as compared to control samples, but significant reduction was observed under the pathogen-challenged condition. The chlorophyll content was highest in the *T. harzianum* + *T. asperellum*-treated sample (33.3 mg·g^−1^ at 48 h), followed by the *T. asperellum*-treated (26.56 mg·g^−1^), *T. harzianum*-treated (26.25 mg·g^−1^), and *P. dendritiformis*-treated (19.33 mg·g^−1^) chilli fruits, and it was lowest in the pathogen-inoculated sample (7.18 mg·g^−1^) as compared to control samples. Similarly, the chlorophyll b content was highest in the *T. harzianum* + *T. asperellum* (37.30 mg·g^−1^), followed by the *T. harzianum* (34.42 mg·g^−1^), *T. asperellum* (29.49 mg·g^−1^), *P. dendritiformis* (21.64 mg·g^−1^), and it was lowest in the pathogen-challenged sample (6.83 mg·g^−1^). The total chlorophyll content was reduced in the pathogen-inoculated sample (12.07 mg·g^−1^) and greatest in the *T. harzianum* + *T. asperellum*-treated sample (46.15 mg·g^−1^), followed by the *T. harzianum*-treated (42.36 mg·g^−1^), *T. asperellum*-treated (38.42 mg·g^−1^), and *P. dendritiformis*-treated (31.52 mg·g^−1^) chilli fruits. The development of anthracnose lesions on the chilli fruits was least in the *T. harzianum* + *T. asperellum-* followed by *T. harzianum-*, *T. asperellum-*, and *P. dendritiformis*-treated samples and highest in the pathogen-inoculated chilli fruit (Figure 9).

### 3.8. Defense Enzyme Assay

#### 3.8.1. Peroxidase (POX) Activity

The chilli seeds primed with *P. dendritiformis, T. harzianum, T. asperellum,* and *T. harzianum* + *T. asperellum* enhanced the peroxidase activity from 0 to 48 h after pathogen inoculation in respect to pathogen inoculation and the untreated and unchallenged (control) samples. The peroxidase activity was highest in *T. harzianum* + *T. asperellum*-treated samples, followed by *T. harzianum*-, *T. asperellum*-, *P. dendritiformis*-treated samples and pathogen-inoculated samples. The peroxidase activity was least in the untreated (control) and unchallenged plant. In peroxidase enzymatic assay, the chilli seeds bioprimed with *T. harzianum* + *T. asperellum* and *T. harzianum* showed maximum activity at 72 hpi, while *P. dendritiformis’* maximum activity was observed at 48 hpi. In these results, bioprimed seeds with *T. harzianum* + *T. asperellum* (57.28 unit) under pathogen-challenged conditions possessed a significant increase in peroxidase activity up to 1.5-fold as compared to the unprimed pathogen-inoculated samples (39.37 unit).

#### 3.8.2. PAL Activity

The chilli seeds treated with PGPB and PGPFs under a pathogen-challenged conditions significantly increased the PAL activity at a different time interval. Phenylalanine ammonia-lyase activity was maximum in *T. harzianum* + *T. asperellum*-treated samples (at 48 h.p.i), followed by *T. harzianum*-, *T. asperellum*-, *P. dendritiformis*-treated samples under pathogen-challenged condition. The activity of the defence enzyme, PAL, in pathogen-inoculated chilli fruit (21.76 unit) was higher as compared to the untreated and unchallenged (control) samples (9.4 unit). The significant difference of PAL activity between the bioagents treated under pathogen-challenged condition and untreated and challenged samples were increased up to 2-fold (Figure 10).

#### 3.8.3. Polyphenol Oxidase (PPO) Activity

The PPO activity of chilli seeds treated with *P. dendritiformis, T. harzianum, T. asperellum,* and *T. harzianum* + *T. asperellum* under a pathogen-challenged condition, untreated, and *Colletotrichum*-challenged samples are shown in (Figure 11). The chilli seeds treated with bioagents offered a significant enhancement of PPO activity from 0 to 48 hpi with respect to bioagents treated and pathogen challenged. In these results, the activity of PPO was highest in *T. harzianum* + *T. asperellum* (62.96 units at 48 hpi), followed by *T. harzianum* and *T. asperellum.* The *P. dendritiformis*-treated and challenged chilli seeds showed maximum activity of PPO (39.02 units at 72 hpi), followed by pathogen-challenged (27.74 units) and control (12.33 units) samples. The difference in PPO activity between the bioagents treated under pathogen-challenged conditions and untreated and challenged samples was increased up to 1.8-fold.

### 3.9. Antioxidant Enzyme Assay

#### 3.9.1. Superoxide Dismutase (SOD) Activity

The chilli seeds treated with bioagents under a pathogen-challenged condition exhibited a successive enhancement in SOD activity from 0 hpi to 48 h.a.i and subsequently decreased up to 96 h.a.i. The SOD activity was higher in the treated under-challenged-condition samples and the pathogen-inoculated samples at 48 h.a.i. The SOD activity was highest in *T. harzianum* + *T. asperellum* bioprimed seeds (53.81 Units g^−1^ fw), followed by *T. harzianum* (43.45 Units g^−1^ fw) and *T. asperellum* (39.36 Units g^−1^ fw) under a challenged condition and pathogen-inoculated samples (35.95 Units g^−1^ fw) compared to untreated and unchallenged samples. The chilli seeds treated with *P. dendritiformis* offered higher SOD activity than pathogen-inoculated samples at different time intervals.

#### 3.9.2. Catalase (CAT) Activity

In our results, the chilli seeds treated with bioagents under a challenged condition and pathogen-inoculated samples increased the catalase activity from 0 h.p.i to 72 h.a.i. and decreased thereafter. The catalase activity was maximum in *T. harzianum* + *T. asperellum* bioprimed seeds (72.57 unit), followed by *T. harzianum* (68.65 unit), *T. asperellum* (59.82 unit), and *P. dendritiformis* (48.42 unit) under a challenged condition and pathogen-inoculated samples (44.21 unit) compared to the untreated (control) samples. The difference in CAT activity between the bioagents treated under a challenged condition and the pathogen-inoculated samples was increased up to 1.5-fold (Figure 12).

#### 3.9.3. Ascorbate Peroxidase (APX) Activity

The chilli seeds bioprimed with bioagents in under-challenged conditions and pathogen-inoculated samples enhanced the APx activity from 0 h.p.i to 48 h.a.i and decreased subsequently to 96 h.a.i. APx enzyme has a greater affinity for binding with hydrogen peroxide compared to catalase. It detoxifies the reactive oxygen species produced under pathogen attack. The APx activity was maximum in *T. harzianum* + *T. asperellum* bioprimed seeds (83.54 unit), followed by *T. harzianum* (80.42 unit), *T. asperellum* (74.77 unit), and *P. dendritiformis* (73.84 unit) in under-challenged condition and pathogen-inoculated (70.31 unit) samples. The APx was lowest in the untreated (control) and unchallenged samples.

#### 3.9.4. Guaiacol Peroxidase (GPX) Activity

The GPx activity was highest in the treated and pathogen-inoculated samples as compared to the untreated and unchallenged samples at 72 h.p.i and decreased afterwards. A significant increase in GPx activity was observed between the bioagents (PGPB and PGPFs) in the treated and control samples at 72 h after pathogen inoculation. The GPx activity was maximum in *T. harzianum* + *T. asperellum* bioprimed seeds (83.32 Units mg^−1^ protein), followed by *T. harzianum* (76.58 Units mg^−1^ protein), *T. asperellum* (72.70 Units mg^−1^ protein), and *P. dendritiformis* (64.72 Units mg^−1^ protein) in under-challenged condition and pathogen-inoculated (57.46 Units mg^−1^ protein) samples. It was observed that the GPx activity was lowest in the untreated and unchallenged (control) samples (Figure 13).

#### 3.9.5. Estimation of Total Phenolic Content

The total phenolic content in chilli fruit treated with bioagents in under-challenged conditions and pathogen-inoculated samples increased consistently from 0 to 48 h.a.i. A significant difference in total phenolic content was increased in bioagents treated as compared to control samples at 48 h after pathogen inoculation. The phenolic content was maximum in *T. harzianum* + *T. asperellum* (509.29 unit), followed by *T. harzianum* (457.78 unit), *T. asperellum* (421.13 unit), *P. dendritiformis* (384.33 unit) and pathogen-inoculated samples (354.33 unit) as compared to the untreated and unchallenged (control) samples (Figure 14).

### 3.10. Histochemical Studies

#### 3.10.1. Detection of Hydrogen Peroxide Accumulation in Leaves

The production of the reactive oxygen species, hydrogen peroxide, has been observed as a dark brown stain by using 3,3′-diaminobenzidine as a reactant to react with H_2_O_2_ in the bioagents treated, challenged, and pathogen-inoculated leaves. The accumulation of H_2_O_2_ was found to be increased with successive development of anthracnose symptoms and became more prominent in pathogen-inoculated leaves at 48 h.a.i. The detection of H_2_O_2_ accumulation was highest in pathogen-inoculated samples followed by *P. dendritiformis*, *T. asperellum*, *T. harzianum,* and *T. asperellum* + *T. harzianum* treated under-challenged-condition samples. The dark brown stain was completely absent in the untreated and unchallenged (control) samples.

#### 3.10.2. Detection of O_2_^−^ Deposition in Leaves

Superoxide anion (O_2_^−^) accumulation was visualized as a dark blue colouration of formazan formed by the reduction of the nitroblue tetrazolium compound. The formazan compound’s dark blue colouration was more prominent in pathogen-inoculated leaves at 48 h.a.i., followed by bioagents treated under-challenged-condition samples. The deposition of superoxide anion was highest in pathogen-inoculated samples, followed by the *P. dendritiformis*, *T. asperellum*, *T. harzianum,* and *T. asperellum* + *T. harzianum* treated samples under a challenged condition. The dark blue formazan compound was absent in untreated and unchallenged (control) samples (Figure 15).

## 4. Discussion

The rhizospheric and phyllospheric microbes (PGPB and PGPF) are involved in vegetative growth and development and also reduce the pathogen invasion through the soil and foliar attack. These PGPB and PGPF are well known to prevent the entry of phytopathogens by strengthening the mechanical tissue (cell wall, callose, and lignin deposition), inducing the production of defence-related enzymes and ROS molecules, which ultimately lead to systemic resistance [35,55]. Due to their useful aspects, these microbes are commercially exploited as biofertilizers and biopesticides that are sustainable for the agricultural field as compared to chemical practices. In this study, *P. dendritiformis*, *T. asperellum,* and *T. harzianum* isolates were assessed for their biocontrol activity against *C. truncatum*, which causes anthracnose disease of chilli. The BCAs have the potential to produce secondary metabolites and enzymes that inhibit the penetration, proliferation, and establishment of pathogens inside the plants [56]. Anthracnose is a destructive disease in chilli fruit caused by *C. truncatum*. To control this pathogen, we isolated a potent, *P. dendritiformis,* from the root of chilli, which is efficient in inhibiting the radial growth of *C. truncatum*. The percent of radial growth inhibition began after 4 days of incubation and introduced up to the stationary phase. In dual culture assay, the zone of interaction revealed that *P. dendritiformis* inhibit the radial growth by lysis, disintegration, and perforation of mycelium, and ultimately the leakage of cytoplasmic contents leading to cell death in *C. truncatum* hyphae.

In the present study, the chilli seeds that were primed with *P. dendritiformis*, *T. asperellum,* and *T. harzianum* showed a significant reduction in lesion development and disease severity index in the in vivo pathogenicity test. The results revealed that the development of induced systemic resistance and reduction in fruit lesions was higher in *T. asperellum* + *T. harzianum* treated compared to independent-treated and *C. truncatum*-challenged samples. Also, the chilli seeds primed with *T. asperellum* + *T. harzianum* showed maximal disease protection of 78.67%, followed by the *T. harzianum* (70%), *T. asperellum* (64%), and *P. dendritiformis* (56%) in under-challenged-condition samples, whereas *C. truncatum*-infected chilli fruits (6%) served as control. Wei et al. [57] reported that seeds that were treated with PGPR suppressed the *C. orbiculare*, the causal agent of cucumber anthracnose, and enhanced the vegetative growth parameters and defence enzymes. Similar results were confirmed in our study, as seeds primed with *P. dendritiformis*, *T. harzianum*, *T. asperellum,* and *T. asperellum* + *T. harzianum* prevented the ingression of *C. truncatum* in chilli fruit and increased the production of antioxidative and defence enzymes.

In our results, the maximum increase in chlorophyll content (chlorophyll a and chlorophyll b and total chlorophyll content) was observed in *T. asperellum + T. harzianum*-treated seeds, followed by *T. harzianum, T. asperellum, P. dendritiformis* under *C. truncatum*-challenged and pathogen-infected plants as compared to the untreated and unchallenged (control) samples. In the pathogen-infected plant, the chlorophyll content was at the maximum at 48 h.p.i and subsequently decreased due to pathogen ingression, proliferation, and establishment inside the host plant led to damage of photosynthetic pigments and eventually cell death followed.

The production of reactive oxygen species (H_2_O_2_ and O_2_^−^) indicates the chilli–*C. truncatum* incompatibility reaction at 48 hpi under the greenhouse condition. The generation of ROS molecules in the host–pathogen interaction led to the synthesis of various defence enzymes, such as peroxidase, polyphenol oxidase, and phenylalanine ammonia lyase, and antioxidative enzymes, such as superoxide dismutase, catalase, ascorbate peroxidase and guaiacol peroxidase (Figure 16) [58,59]. In our study, the bioprimed seeds enhanced the activities of peroxidase, polyphenol oxidase, and phenylalanine ammonia lyase at 48 h and 72 h after *C. truncatum* inoculation. The activities of POX, PAL, and PO were highest in *T. asperellum* + *T. harzianum*-treated samples, followed by the *T. harzianum*, *T. asperellum*, and *P. dendritiformis* under-challenged-condition samples and *C. truncatum*-infected samples as compared to the control samples. The results revealed that chilli fruits challenged with *C. truncatum* significantly enhanced the activities of superoxide dismutase, catalase, ascorbate peroxidase, and guaiacol peroxidase at 24 h and 48 h as compared to the untreated and unchallenged (control) samples, but the maximum antioxidative enzyme activities were observed in the combination of *T. asperellum* + *T. harzianum* under *C. truncatum*-challenged condition. SOD is an antioxidative enzyme that dismutates the superoxide anion (O_2_^−^) into hydrogen peroxide and molecular oxygen, and these harmful by-products are converted into H_2_O and O_2_^−^ by various antioxidative enzymes, such as catalase, ascorbate peroxidase, and guaiacol peroxidase [60].

The oxidative burst or rapid production of H_2_O_2_ and O_2_^−^ indicated the early defence response in plants against various phytopathogens [61]. The above result was corroborated by histochemical analysis of DAB and NBT staining in chilli leaves against the *C. truncatum* infection. The results of the histochemical studies showed that the accumulation of H_2_O_2_ and O_2_^−^ indicates the deposition of dark brown and blue colouration on the leaf’s surface at 48 h after *C. trunatum* inoculation. These dark brown and blue diformazan precipitates were more eminent in pathogen-challenged leaves, followed by the *P. dendritiformis-*, *T. asperellum*-, *T. harzianum-,* and *T. asperellum* + *T. harzianum*-treated samples upon pathogen-challenged condition as compared to the untreated and unchallenged (control) condition, respectively. At 48 hpi, there was a host–pathogen incompatibility reaction that resulted in more accumulation of ROS. However, after 48 hpi, the accumulation of ROS was significantly reduced due to development of anthracnose symptoms.

Saxena et al. [62] have reported that the phenolic content was higher in seeds primed with dual consortia of *T. harzianum* and *T. asperellum* under *C. truncatum*-challenged condition as compared to untreated and unchallenged plants (control). A similar result corroborated in our study as the total phenolic content enhanced upon seed priming that led to the strengthening of the mechanical tissue of cell wall (xylem strand) by lignin deposition that induced systemic resistance in chilli against the *C. truncatum* ingression. Our results revealed that the generation of ROS, phenol, PAL, PO, POX, SOD, APX, GPx, and CAT enzymes showed the host–pathogen interactions.

## 5. Conclusions

*C. truncatum,* which causes anthracnose disease, is a serious problem in chilli fruit production in India at both red- and post-harvest stage. The conidia of *C. truncatum* spread through water or air and produce a symptom containing dark black water-soaked sunken lesions with the concentric ring of acervuli on the chilli fruit. After the establishment of anthracnose disease, the pathogen inocula destroy the whole chilli crop field. To control the anthracnose disease, the use of BCAs reduces the chemical load, which is unsustainable to the agriculture field. Our results conclude that plants treated with *P. dendritiformis*, *T. harzianum,* and *T. asperellum* possess enormous potential to induce systemic resistance (ISR) against *C. truncatum*. These BCAs not only induce the defence system but also suppress the pathogen establishment and ultimately reduce the anthracnose disease’s development in chilli. Further, work on the development of biofertilizer using *P. dendritiformis* for the control of fruit rot under natural conditions and for use as a bioinoculant.

## Figures and Tables

**Figure 1 jof-07-00307-f001:**
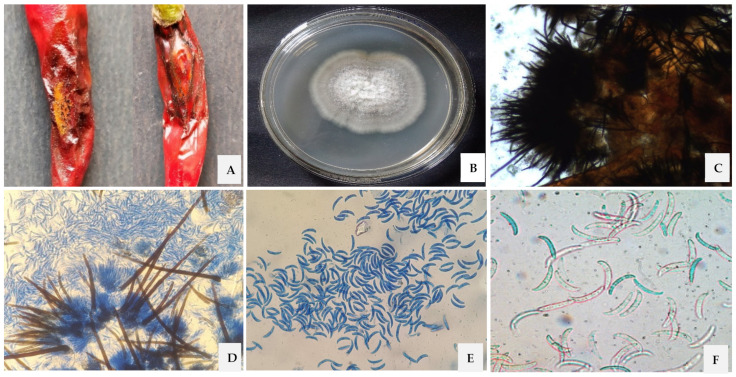
(**A**) Chilli fruit associated with anthracnose disease. (**B**) Pure colony *C. truncatum* after 5 days on PDA medium. (**C**) Acervuli on chilli fruit. (**D**) Conidia on conidiophore and setae. (**E**) Conidia of *C. truncatum* at 10×. (**F**) Sickle-shaped conidia of *C. truncatum* at 40×.

**Figure 2 jof-07-00307-f002:**
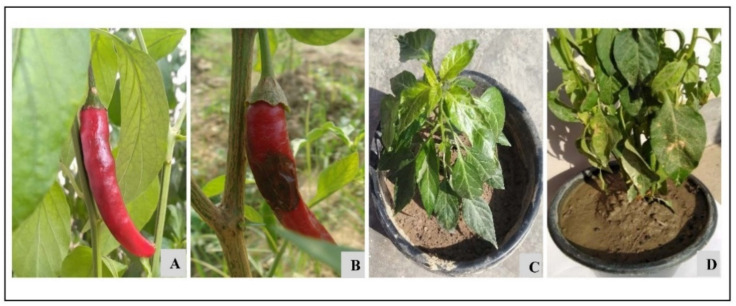
Pathogenicity test on chilli fruit and leaf. (**A**) Water-treated control showing no symptoms on the fruit of *Capsicum annuum* (cv. Surajmukhi). (**B**) Fruit challenged with the conidial suspension of *C. truncatum* showing anthracnose symptoms after 7 days of inoculation. (**C**) Healthy plant served as a control. (**D**) Leaves sprayed with the conidial suspension showing symptoms after 7 days of inoculation.

**Figure 3 jof-07-00307-f003:**
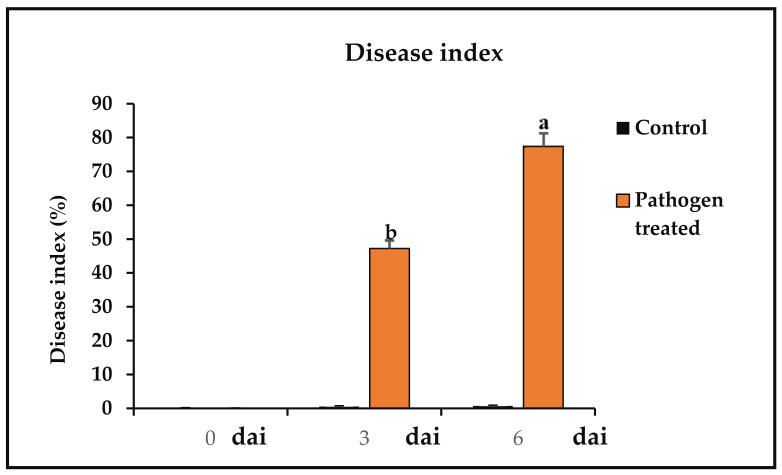
The disease index of *C. truncatum*-inoculated chilli fruit. Each value is expressed as mean of triplicates, and the bars sharing the same alphabetical letters are not significantly different (*p* ≤ 0.05) using Duncan’s multiple range test. The vertical bar designates the standard error.

**Figure 4 jof-07-00307-f004:**
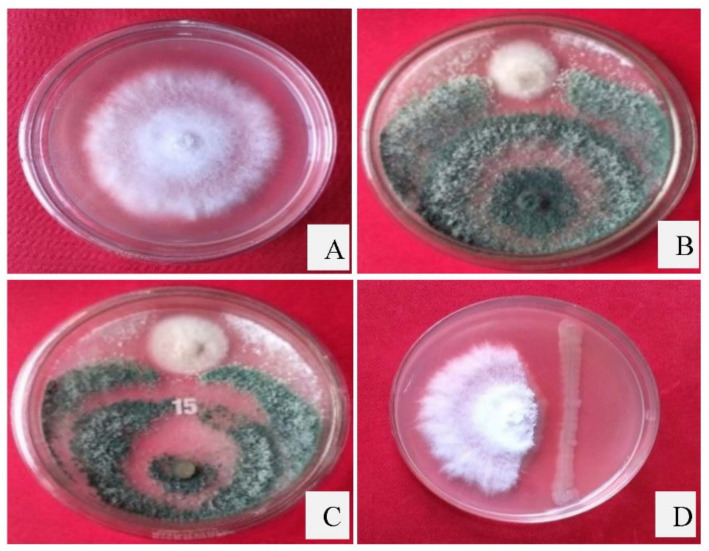
The effects of *T. harzianum, T. asperellum,* and *P. dendritiformis* on the inhibition of radial growth of *C. truncatum*. (**A**) Culture plate of *C. truncatum* serving as control; (**B**) Antagonistic activity by *T. harzianum* after 6 days post-inoculation; (**C**) *T. asperellum* after 6 DPI; (**D**) *P. dendritiformis* after 6 DPI.

**Figure 5 jof-07-00307-f005:**
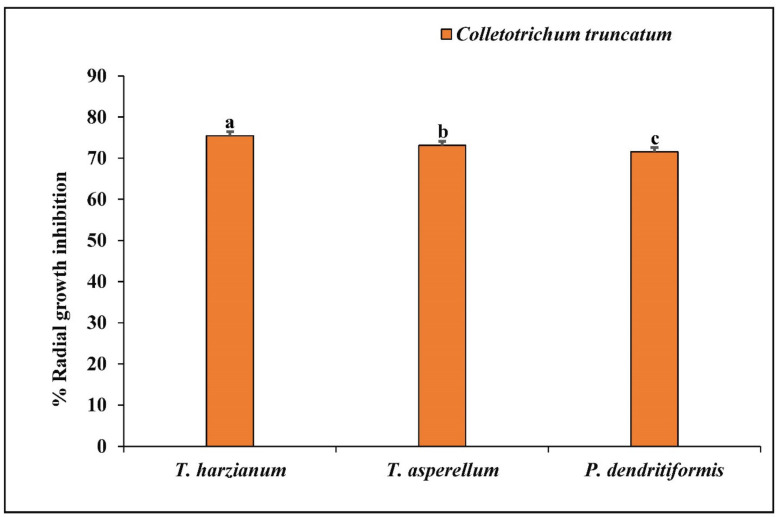
Effects of *T. harzianum*, *T. asperellum,* and *P. dendritiformis* on the inhibition of hyphal growth of *C. truncatum*. Each value is expressed as the mean of triplicates, and the bars sharing the same alphabetical letters are not significantly different (*p* ≤ 0.05) using Duncan’s multiple range test. The vertical bar designates the standard error.

**Figure 6 jof-07-00307-f006:**
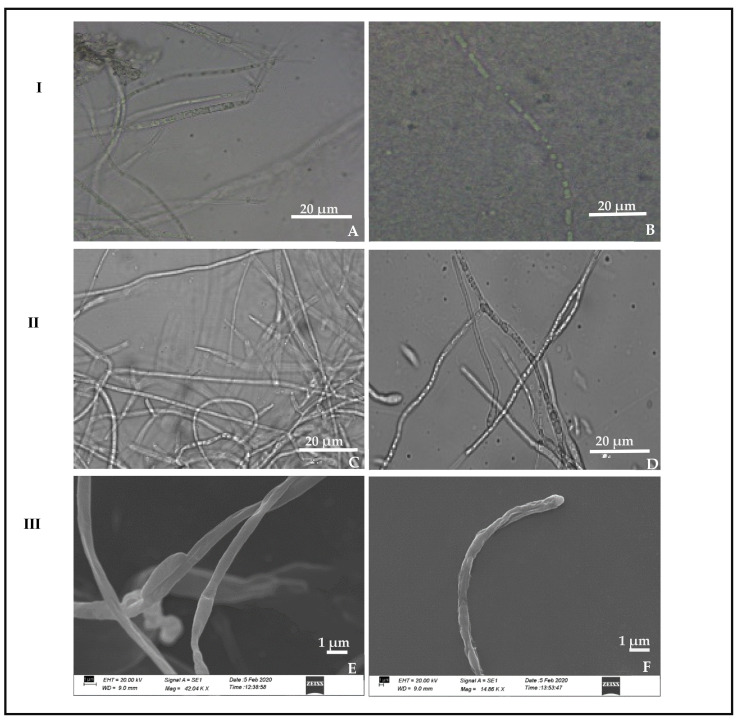
Antagonistic activity of *P. dendritiformis* in a dual culture plate. (**I**) Light microscope; (**A**) normal hyphae, (**B**) *P. dendritiformis* in a dual culture plate showing hyphal degradation and lysis. (**II**) Bright field; (**C**) normal hyphae, (**D**) hyphal swelling, disintegration, and leakage of cytoplasmic contents. (**III**) Scanning electron microscope; (**E**) normal hyphae, (**F**) lysis, degradation, and disintegration of hyphae.

**Figure 7 jof-07-00307-f007:**
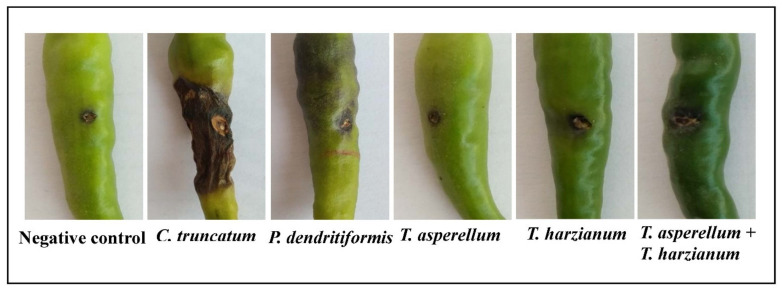
Induced systemic resistance by suppression of anthracnose disease development in chilli fruits treated by *P. dendritiformis*, *T. asperellum*, *T. harzianum,* and *T. asperellum* + *T. harzianum* in challenged and *C. truncatum*-infected samples under greenhouse conditions. The disease lesion development was recorded after ten days post-inoculation of *C. truncatum*. Treatment with bioagents on inoculation of *C. truncatum* showing reduced lesion development as compared to *C. truncatum* inoculation and control.

**Figure 8 jof-07-00307-f008:**
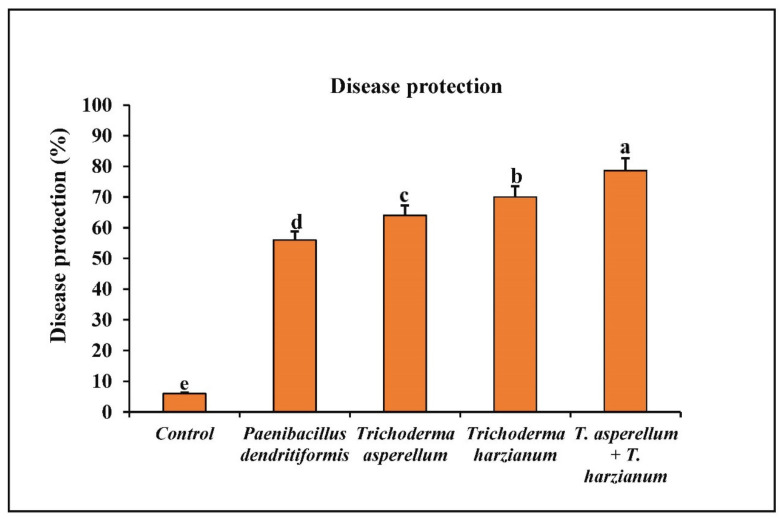
The treatment of chilli with *T. harzianum, T. asperellum,* and *P. dendritiformis* showing protection against anthracnose disease in the greenhouse experiment. Each value is expressed as the mean of triplicates, and the bars sharing the same alphabetical letters are not significantly different (*p* ≤ 0.05) using Duncan’s multiple range test. The vertical bar designates the standard error.

**Figure 9 jof-07-00307-f009:**
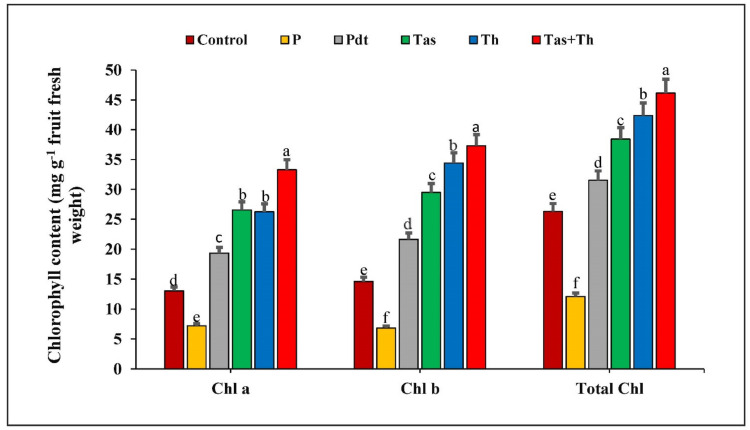
The effect on total chlorophyll, chlorophyll a, and chlorophyll b content in chilli fruits treated with *C. truncatum* and bioagents (Pdt = *P. dendritiformis,* Tas *= T. asperellum,* Th *= T. harzianum,* and Tas+Th = *T. asperellum + T. harzianum*) under a challenged condition and *C. truncatum*-infected samples after 48 h. Each value is expressed as the mean of triplicates, and the bars sharing the same alphabetical letters are not significantly different (*p* ≤ 0.05) using Duncan’s multiple range test. The vertical bar designates the standard error.

**Figure 10 jof-07-00307-f010:**
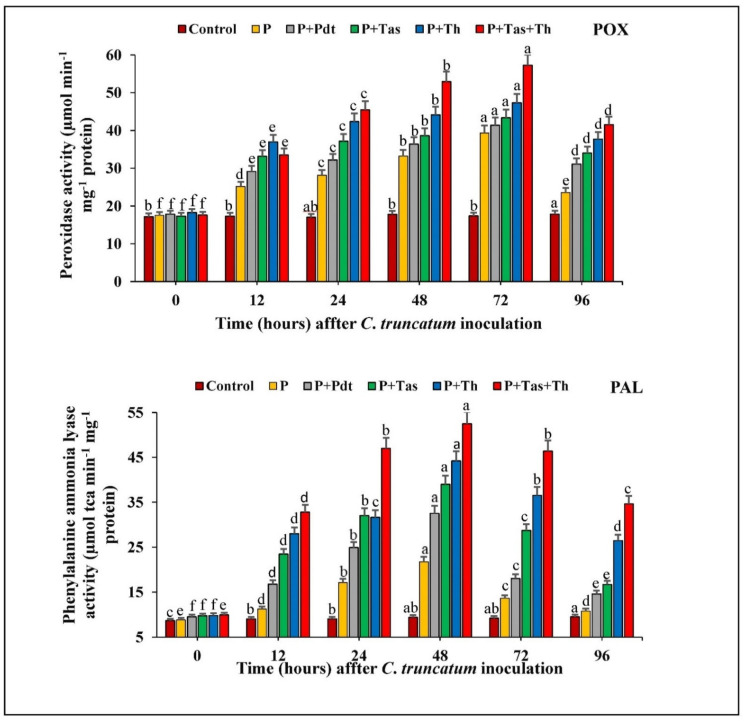
The effect of bioagents (Pdt = *P. dendritiformis,* Tas *= T. asperellum,* Th *= T. harzianum,* and Tas+Th = *T. asperellum + T. harzianum*) on the accumulation of the defence enzymes (POX and PAL) either individually or in the combination of treatment in chilli seeds under a challenged condition and *C. truncatum*-infected samples. Each value is expressed as the mean of triplicates, and the bars sharing the same alphabetical letters are not significantly different (*p* ≤ 0.05) using Duncan’s multiple range test. The vertical bar designates the standard error.

**Figure 11 jof-07-00307-f011:**
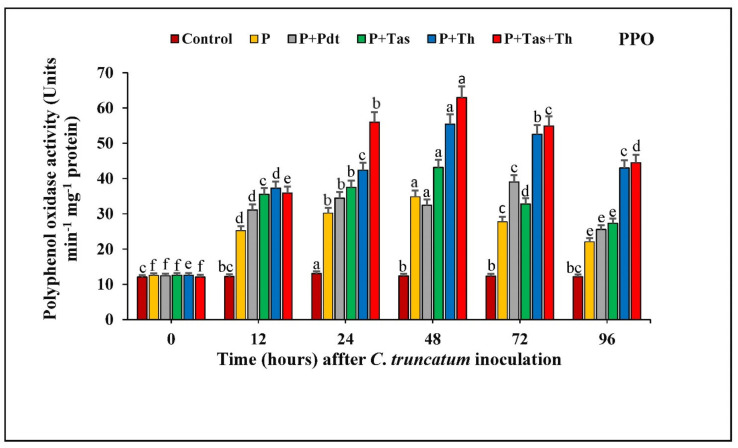
The effect of bioagents (Pdt = *P. dendritiformis,* Tas *= T. asperellum,* Th *= T. harzianum,* and Tas+Th = *T. asperellum + T. harzianum*) on the accumulation of the defence enzyme (PPO) either individually or in the combination of treatment in chilli seeds under a challenged condition and *C. truncatum*-infected samples. Each value is expressed as the mean of triplicates, and the bars sharing the same alphabetical letters are not significantly different (*p* ≤ 0.05) using Duncan’s multiple range test. The vertical bar designates the standard error.

**Figure 12 jof-07-00307-f012:**
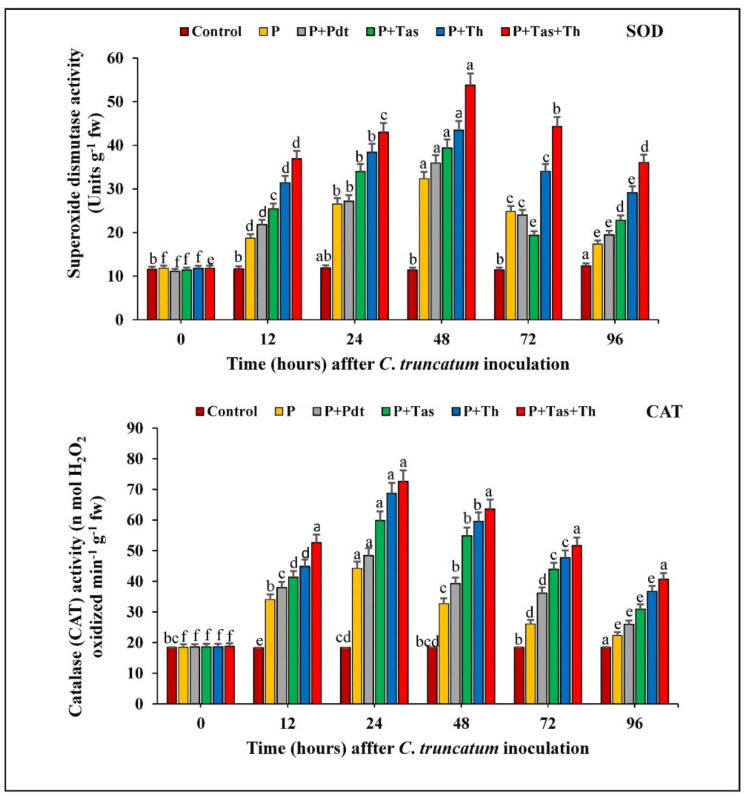
The effect of bioagents (Pdt = *P. dendritiformis,* Tas *= T. asperellum,* Th *= T. harzianum,* and Tas+Th = *T. asperellum + T. harzianum*) on the accumulation of antioxidative enzymes (SOD and CAT) either individually or in the combination of treatment in chilli seeds under a challenged condition and *C. truncatum*-infected samples. Each value is expressed as the mean of triplicates, and the bars sharing the same alphabetical letters are not significantly different (*p* ≤ 0.05) using Duncan’s multiple range test. The vertical bar designates the standard error.

**Figure 13 jof-07-00307-f013:**
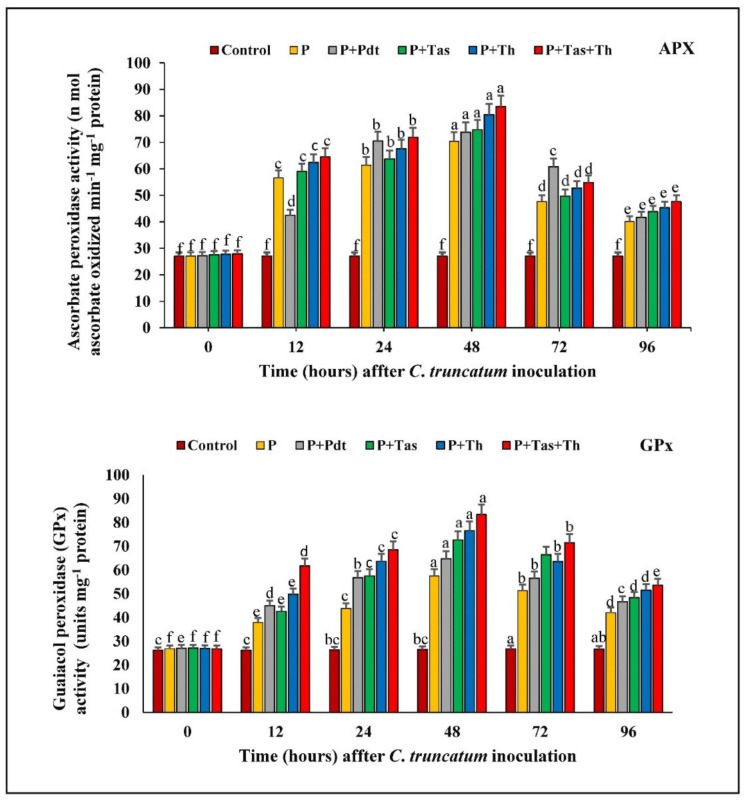
The effect of bioagents (Pdt = *P. dendritiformis,* Tas *= T. asperellum,* Th *= T. harzianum,* and Tas+Th = *T. asperellum + T. harzianum*) on the accumulation of antioxidative enzymes (APx and GPx) either individually or in the combination of treatment in chilli seeds under a challenged condition and *C. truncatum*-infected samples. Each value is expressed as the mean of triplicates, and the bars sharing the same alphabetical letters are not significantly different (*p* ≤ 0.05) using Duncan’s multiple range test. The vertical bar designates the standard error.

**Figure 14 jof-07-00307-f014:**
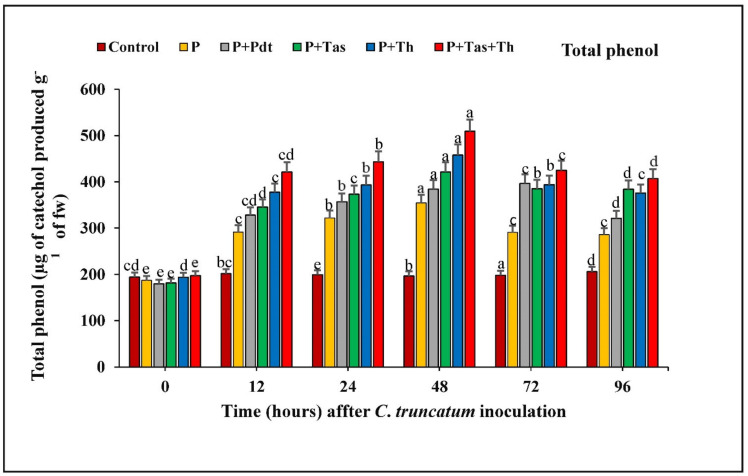
The effect of bioagents (Pdt = *P. dendritiformis,* Tas *= T. asperellum,* Th *= T. harzianum,* and Tas+Th = *T. asperellum + T. harzianum*) on the production of total phenols either individually or in the combination of treatment in chilli seeds under a challenged condition and *C. truncatum*-infected samples. Each value is expressed as the mean of triplicates, and the bars sharing the same alphabetical letters are not significantly different (*p* ≤ 0.05) using Duncan’s multiple range test. The vertical bar designates the standard error.

**Figure 15 jof-07-00307-f015:**
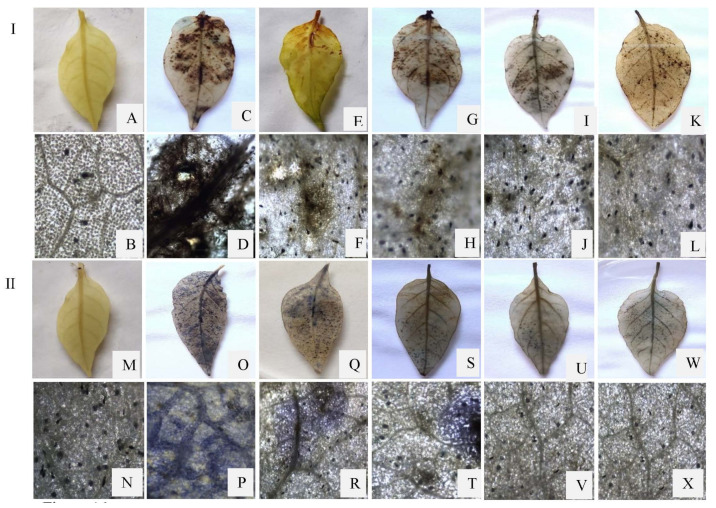
(**I**) Accumulation of H_2_O_2_ in chilli leaves as visualized by 3,3′-diaminobenzidine staining at 48 hpi. (**A**,**B**) Untreated and unchallenged (control); (**C**,**D**) challenged with a pathogen; (**E**,**F**) Pre-treated with *P. dendritiformis;* (**G**,**H**) Pre-treated with *T. asperellum*; (**I**,**J**) Pre-treated with *T. harzianum*; (**K**,**L**) Pre-treated with *T. asperellum* + *T. harzianum*. (**II**) Superoxide anion production in chilli leaves as visualized by nitroblue tetrazolium staining at 48 hpi. (**M**,**N**) Untreated and unchallenged (control); (**O**,**P**) challenged with a pathogen; (**Q**,**R**) Pre-treated with *P. dendritiformis*; (**S**,**T**) Pre-treated with *T. asperellum*; (**U**,**V**) Pre-treated with *T. harzianum*; (**W**,**X**) Pre-treated with *T. asperellum* + *T. harzianum*.

**Figure 16 jof-07-00307-f016:**
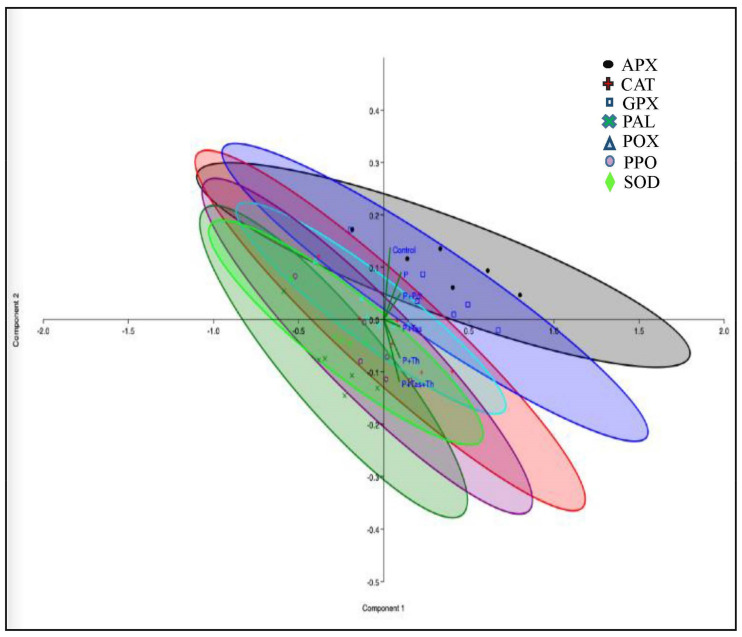
Principal component analysis (PCA) of chilli fruits treated with bioagents under pathogen-challenged, untreated and unchallenged as negative control, and untreated and pathogen-challenged condition as positive control were harvested at different time intervals like 0, 12, 24, 48, 72, and 96 hpi.

**Table 1 jof-07-00307-t001:** The effects of *T. harzianum*, *T. asperellum,* and *P. dendritiformis* on the percent inhibition of radial growth of *C. truncatum* (values are the average of three replicates ± SEM).

Antagonists	Percent Inhibition of Radial Growth
*Trichoderma harzianum*	75.46 ± 0.45
*Trichoderma asperellum*	73.09 ± 0.56
*Paenibacillus dendritiformis*	71.56 ± 0.30

## Data Availability

The DNA sequence obtained from this study have been submitted to the GenBank database with NCBI accession numbers; MW541903 and MW365553.

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
