# Peer review of "Systemic Resistance in Chilli Pepper against Anthracnose (Caused by Colletotrichum truncatum) Induced by Trichoderma harzianum, Trichoderma asperellum and Paenibacillus dendritiformis"

_jof, 2021, doi:10.3390/jof7040307_

Round 1

Reviewer 1 Report

I have only read the abstract and I found so many inconsistencies that I am not continuing reading the rest of the work. The experimental design is not adequately explained and nearly every sentence is incomprehensible to me. In many cases I try to guess what the authors mean, but far too often I fail to reach a conclusion and I am afraid I am not willing to continue reading the manuscript. I recommend substantial review both of the scientific message and of English usage. Here are a few examples:

L9 – Remove ‘the’ before Paenibacillus

L10 – induce

L11 – the scientific name should be given after the common name and not in-between the two words of the common name

L11 – remove comma

L12 – Start sentence using Bright field…

L13 – Is it “C. truncatum against P. dendritiformis” or the other way round? Why italics in “in dual plate assay”?

L14 – Seeds are soaked in soil? The soil is soaked in biocontrol agents (suspension)? This is quite unclear.

L14-15 – Chilli seeds were pretreated or Chilli seeds pretreated (…) induced?

L16 – I do not understand the use of a verb in an acronym. Also, this acronym is not used in the rest of the manuscript.

L17 – You already said that anthracnose is caused by C. truncatum.

L18-19 – What is “Trichoderma asperellum + Trichoderma”?

L20 – I presume that ‘unchallenged’ refers to plants not inoculated with C. truncatum. In that case, how come disease index is reduced in comparison to unchallenged plants? Was there any disease on negative controls?

L21 – What are BCAs? What is biopriming? What does the value 78.67% stand for?

L22 – What are these values?

L23 – What does 6% in control represent? What is control here?

L23 – Please do not put a comma between noun and verb.

L29-30 – “The accumulation of (…) were [was] maximum at 48 hpi followed by Paenibacillus dendritiformis (…)” ?? How come after 48 hpi comes Paenibacillus dendritiformis?

Author Response

Dear reviewer,

We went through your constructive suggestion and improved the MS subsequently making it more reader's friendly. Further, all the errors have been removed and the same has been attached for your perusal. 

Q. Seeds are soaked in soil? The soil is soaked in biocontrol agents (suspension)? This is quite unclear.

Response: The seeds were first of all coated with CMC (carboxymethyl cellulose) and dried. Afterwards, the coated seeds were soaked in inoculum suspension of biocontrol agents. 

Q. What is “Trichoderma asperellum + Trichoderma”?

Response: This stands for consortia developed by mixing conidial suspension of two different species of Trichoderma for inducing systemic resistance in chilli fruits.

Q. 20.

Response : Your presumption is wrong there is no such thing as for as I am  concerned unchallenged plant have zero disease index  further there is none of my treatment which have reduced disease index in comparison to unchallenged plant 

L21 – What are BCAs? What is biopriming? What does the value 78.67% stand for?

Respponse: BCA means biocontrol agents; Seeds coated with biocontrol agents; 78.67 is the disease protection value showed by mentioned consortium   

L23 – What does 6% in control represent? What is control here?

Response: 6 % control represent only 6% protection of chilli fruit area. Here control represent the C. truncatum challenged chilli fruit. Here untreated and unchallenged served as control mentioned in disease protection methodology

Reviewer 2 Report

what is the aim of the article? I did not find out clear aim .

keywords should be different than the paper title.

the comments are provided in the text.

some English language correction are needed.

119 line: have you confirmed by PCR the Colletotrichum isolates? How many isolates used in this study?

133 line: how was chilli grown? open field, gashouse? pots? What cultivar?

139 line: have you confirmed the species PCR?

142 line: all 13 were the same bacteria?

164-170 line: reference

188 line: no info from were obtained Trichoderma strains.

190 line: as I understand you grew the chilli for the pathogenicity. so please add first sentence the purpose for this experiment.

197 line: how and where the leaves, petri? Filter paper? Methods add. As from results I see that leaves were inoculated on plant. Add detail info in methods. In which part of leave you inoculated?

202-207 line: as I understand you inoculated chilli fruits on plant growing. add details were the plants grown.

208-212 line: reference

The methodology should be structured more precise, because now in the end we find information, which we are looking in front.

222 line: one treatment I understand are 3 plates. but how many times repeated the experiment,.

254 line: 9 plants per treatment? repeated how many times?

272 line: add BBCH it would be easier to follow.

Overall provided data should be used for 2 separate publication.

314 line: whole section how many analytical replicates? how many times repeated the analysis?

453 line: no info of disease index in methods. Formula. Also for which treatment?

473 line: no info what are A, B in this figure

478 line: how radial growth inhibition measured? formula? Add methods

Have you tested in petri the effect on anthracnose of T. asperellum+ T.harzianum ?

Figure 10-11-12-13-14-15: add treatments description , P, Pdt, Tas .. and so on,. Please after figure add description of abbreviations

746 line: how Percent inhibition of radial growth  was measured? add info methods part

Author Response

Dear reviewer,

We have followed all your constructive instructions/suggestion and the revised version of the MS has been uploaded for your perusal. In addition, we are also submitting a proper reply to all the quarries raised.  

Round 2

Reviewer 1 Report

I list several issues that must be considered by the authors. Among them, two are of high relevance. One is that the inoculation experiments using injured leaves and injecting spores into fruits are not appropriate to claim pathogenicity. I thing that the work on the antagonistic role is still valid, but those inoculation methods are not adequate to show pathogenicity. Also, Figure 3 is of poor quality, not only concerning the images themselves, but their interpretation and indeed the methods used for obtaining them. Figure 3 must be deleted. This does not affect the overall message.

L2-3 “anthracnose disease” is redundant - please delete “disease”. The word “induced” should be followed by the names of the inducers. Being this the Journal of Fungi, I consider most relevant to include the name of the fungal pathogen in the title. I thus suggest the title to be: “Systemic resistance in chilli pepper against anthracnose (caused by Colletotrichum truncatum) induced by Trichoderma harzianum, Trichoderma asperellum and Paenibacillus dendritiformis”

L11 Again, “anthracnose disease” is redundant - please delete “disease”.

L13 Is it “C. truncatum against P. dendritiformis” or the opposite?

L19 To make it clear that the percentage values account for a measure of disease protection, this part of the sentence would better read “seed primed with T. asperellum + T. harzianum showed maximum disease protection (78.67 % disease protection)”.

L22 Remove comma after BCAs. Against C. truncatum (remove “the”).

L26 The authors detected, not the stains. The stains were used to detect…

L27-29 The authors performed some changes in this sentence but I still cannot understand it. To my comprehension, it states that at 48 hpi the accumulation of ROS was maximum. After that, what happens? The authors state that after that follows “P. dendritiformis, T. asperellum, T.  harzianum and T. asperellum + T. harzianum treated tissue upon C. truncatum challenged condition as compared to the control.” Please rephrase this as I cannot understand what you mean!

L31 As biocontrol agents. Plural.

L32 in preventing infection by C. truncatum

L51-52 What does the “respectively” apply to? Why two different types of measurements? What are lakh tonnes?

L52-53 What is the subject of this sentence? What does “their” refer to? Who needs to maintain the fruit yield?

L53-54 Please provide a reference to the yield losses in recent years.

L56 Please remove “etc”.

L57 Please include also the classifier of C. truncatum.

L57-58 At this point the authors are stressing the relevance of the disease and thus I suggest that they simplify the sentence by saying “Among these diseases, anthracnose of chilli alone caused 50% yield loss worldwide [6,7].” Then in the next paragraph you can detail the identity of the pathogen. The name C. capsici was considered a synonym of C. truncatum, but that does not necessarily imply that C. truncatum as we now know it was previously C. capsici. Other fungi that never were under the name C. capsici are currently under C. truncatum.

L60-63 There are up to 32 species of Colletotrichum associated to anthracnose in Capsicum spp. In fact, Colletotrichum truncatum and C. scovillei are reported as the prevalent pathogens in Asia (Silva et al. 2019; https://doi.org/10.1186/s43008-019-0001-y), but the authors should account for the high diversity of pathogens at this point and recognise that the disease is nor caused by C. truncatum alone.

L81-82 Trichoderma spp. are soil inhabitants and have

L84 antibiotics

L85 Please consider competition instead of competence

L87-88 It is necessary to give the definition of these acronyms, even if they were already defined in the abstract

L90 via increasing the activity or via increases in the activity

L91 What does PO stand for? This is not a good acronym, as it is impossible to look for all “po” occurrences…

L96 the acronym PR is defined here but it was used before (e.g., L88)

L98 lead (the subject of the sentence is plural)

L99-100 I cannot find a verb in this sentence. Could it be “The biochemical defense response is” (instead of “as”)?

L105-106 Several reports (…) have been studied”? Perhaps “have been made”.

L108 by employment of its

L112 replace “against” by “caused by”

L118 Why do you have molecular characterization here and molecular identification in 2.2? There is nothing about molecular characterization in 2.1. I suggest the title to be “Isolation and morphological and cultural identification of the pathogen”

L119 How do you know beforehand that the fungus belonged to the species C. truncatum? At this point you can say “The fungal pathogen was isolated”. Also, what are samples associated with anthracnose. I suggest “samples with anthracnose”. And please remove “disease”, as “anthracnose disease” is a pleonasm.

L125 Same concentration for both antibiotics? It is not clear.

L138 autoclaved

L141 mixed thoroughly

L142 Please check your protocol: the purpose of the phenol:chloroform step is not DNA precipitation.

L144 Now you are precipitating DNA.

L148 1× Tris- EDTA means nothing. Please give concentrations of both reagents and pH. Also, I guess you check the quality of the extracted DNA and quantified it, before going for PCR. Please give some information on that.

L150-153 These two sentences basically say the same. I suggest retaining the first sentence but adding the reference [29].

L154 containing 20−50 ng

L155 dNTPs. Please give details of the manufacturer of Taq DNA polymerase and buffer.

L155-157 How many cycles?

L172 Please give more details and references concerning bacterial identification. Bacteria usually require biochemical (and molecular) tests for identification, not just morphology.

L178 and was selected and used for further experiments

L180 Weren’t the bacteria identified morphologically as described in 2.3.1? Perhaps in 2.3.1 they were only differentiated but not identified.

L182 And what about the methods for DNA isolation and amplification? There is dramatically less details here than those given for identification of the fungal pathogen.

L186 This expression “BHU BOT RYRL4” is the fungus reference. However, KR856210 is not the fungus accession in NCBI. It is the reference for the ITS sequence of “BHU BOT RYRL4”. It is not the same! The same goes for BHU BOT RYRL1.

L189 Seven days old

L206 Seeds of the susceptible (…) were obtained

L207 In this case the purpose is not the isolation of microorganisms, so I think the disinfection protocol should be given again.

L211 pathogenicity of C. truncatum

L213 I don’t think you need to repeat the species and cultivar and the title can be “Pathogenicity test on leaves and fruits”

L214 Why do you need to injure the plants, if this is a pathogen? In that case, this is not a pathogenicity test…

L221 Same as before. If this is a pathogen, it does not need you to help it into the fruit. Otherwise it is not a pathogenicity test, it is just testing its opportunistic behaviour.

L231 Remove “(1x106 conidia mL-1) on chilli fruit with the help of a sterile syringe”.

L301 This title needs more details

L317 biocontrol

L318 At the end, 120 days old plants

L370 Sodium phosphate buffers typically contain different quantities of Na2HPO4 and NaH2PO4 (and it is the equilibrium established between them that makes the buffer effect) rather than containing Na3PO4. Please check.

L468 Image B in Figure 1 is distorted.

L472 What if you had inoculated spores onto intact leaves and fruits? That would be a pathogenicity test. They way this inoculation was performed, I do not think the authors can claim pathogenicity.

L487 I do not think that this figure has quality for being published. I do not understand what are germinating hyphae. Germinating from what and towards what? Figure B is imperceptible. Figure C suggests an appressorium inside the cuticle at 72h. Appressoria form on the surface and typically within 18-24h after inoculation. The method described in 2.7.3 is not adequate. For this, transversal sections of the leaves should have been obtained. Figure 3 should be deleted or otherwise these experiments repeated.

L514 Panels A, B and C in Figure 5 are deformed. “on the percent inhibition” – remove percent, as there are no percentages in the image. “served as control” – I suggest “serving as control”.

L523 There is no need for the inset saying “Colletotrichum truncatum”, as there is no other parameter in this graphic.

L533 No scales are given in Figure 7.

Author Response

Dear reviewer,

Thank you for your constructive suggestions. We followed all your valuable inputs and improved our MS accordingly. The same has been uploaded for your consideration.

The response for some of your major quarries are as follows:- 

Response to the quarries raised by reviewer 1:-

Q. I list several issues that must be considered by the authors. Among them, two are of high relevance. One is that the inoculation experiments using injured leaves and injecting spores into fruits are not appropriate to claim pathogenicity. I thing that the work on the antagonistic role is still valid, but those inoculation methods are not adequate to show pathogenicity. Also, Figure 3 is of poor quality, not only concerning the images themselves, but their interpretation and indeed the methods used for obtaining them. Figure 3 must be deleted. This does not affect the overall message.

Response: We followed the pin-prick inoculation method for testing the pathogenicity on fruits. The same method was used by several investigators in their research. For perusal please check the following references:- De Silva, D.D., Ades, P.K., Crous, P.W., Taylor, P.W.J., 2017. Colletotrichum species associated with chili anthracnose in Australia. Plant Pathol. 66, 254–267.

Dela Cueva FM, Mendoza JS, Balendres MA. 2018. A new Colletotrichum species causing anthracnose of chilli in the Philippines and its pathogenicity to chilli cultivar Django. Crop Protection 112: 264-268.

   However, for testing the pathogenicity on leaves, we directly sprayed the conidial suspension of the pathogen on leaves by following the method of  Chakraborty et al. (2019).

Chakraborty N, Mukherjee K, Sarkar A, Acharya K. Interaction between Bean and Colletotrichum gloeosporioides: Understanding Through a Biochemical Approach. Plants (Basel). 2019; 8(9): 345.

Further, if their is any flaw in the MS regarding the methodology, we have checked and improved it. Beside this, we also deleted the figure 3 as per your suggestion. 

Q. Is it “C. truncatum against P. dendritiformis” or the opposite?

Response: Yes it is “C. truncatum against P. dendritiformis”.

Q. L27-29 The authors performed some changes in this sentence but I still cannot understand it. To my comprehension, it states that at 48 hpi the accumulation of ROS was maximum. After that, what happens? The authors state that after that follows “P. dendritiformis, T. asperellum, T.  harzianum and T. asperellum + T. harzianum treated tissue upon C. truncatum challenged condition as compared to the control.” Please rephrase this as I cannot understand what you mean!

Response: Histochemical staining states that at 48 hour after pathogen  inoculation (hpi), the ROS accumulation was more prominent on C. truncatum challenged condition followed by P. dendritiformis, T. asperellum, T. harzianum and T. asperellum + T. harzianum treated leaves sample under pathogen challenged condition, respectively. At 48 hpi there was host-pathogen compatibility reaction which resulted in more accumulation of ROS. However, after 48 hpi the accumulation of ROS was significantly reduced due to the development of anthracnose symptom.

Q. L51-52 What does the “respectively” apply to? Why two different types of measurements? What are lakh tonnes?

Response: The phase has been modified as "According to the recent report, the production of dried and green chilli fruits in India has reached around 1.389 million tonnes and 0.0679 million tonnes, cultivated in an area of 0.797 million
hectares [5]."

Q. L52-53 What is the subject of this sentence? What does “their” refer to? Who needs to maintain the fruit yield?

Response: The sentence has been be phrased as "Due to their medicinal effects and daily use, it need to maintain the chilli fruit yield world over." 

Q. L53-54 Please provide a reference to the yield losses in recent years.

Response: Proper references have been incorporated.

Q. L57 Please include also the classifier of C. truncatum.

Response: Proper classifier name has been added.

Q. L60-63 There are up to 32 species of Colletotrichum associated to anthracnose in Capsicum spp. In fact, Colletotrichum truncatum and C. scovillei are reported as the prevalent pathogens in Asia (Silva et al. 2019; https://doi.org/10.1186/s43008-019-0001-y), but the authors should account for the high diversity of pathogens at this point and recognise that the disease is nor caused by C. truncatum alone.

Response: We agree with your suggestions and modified the MS accordingly.  

Q. L118 Why do you have molecular characterization here and molecular identification in 2.2? There is nothing about molecular characterization in 2.1. I suggest the title to be “Isolation and morphological and cultural identification of the pathogen”

Response: Yes. The suggested amendments have been made.

Q. L125 Same concentration for both antibiotics? It is not clear.

Response: We used streptomycin (0.03 g L-1) and chloramphenicol (0.05 g L-1) in PDA medium for isolation. The same have been modified in revised MS. 

Q. L172 Please give more details and references concerning bacterial identification. Bacteria usually require biochemical (and molecular) tests for identification, not just morphology.

Response: Overall, we isolated 13 endophytic bacterial isolates and subjected them for antagonistic activity against the phytopathogen (C. truncatum) using the dual culture method. The endophytic bacterium which showed the maximum antagonistic activity were selected for further studies and identified by using molecular method. As the other bacterial isolates didn't showed the significant antagonistic activity, therefore, we went for their morphological identification only.

Q. L186 This expression “BHU BOT RYRL4” is the fungus reference. However, KR856210 is not the fungus accession in NCBI. It is the reference for the ITS sequence of “BHU BOT RYRL4”. It is not the same! The same goes for BHU BOT RYRL1

Response: All these are NCBI GenBank Accession No. The screenshot of the same have been also uploaded for your consideration.

Reviewer 2 Report

37 line: suggest not use abbreviations in keywords

132 line: add molecular analysis reference or rewrite sentence. I suggest delete this sentence and write in 134 line: Collected isolates in 2.1. were confirmed by molecular analysis.     and you write how extracted DNA.

164 line: suggest in title use full names: Plant growth-promoting bacteria isolates

164 line : add a table of bacteria collected.

167 line: using the method described in 2.1 section.

171 line: bacteria were identified by PCR or not? or only selected for future works?

174 line: add a table of selected bacteria used in this study. You can make one table together with all 13 selected.

You used only one strain of bacteria for future works?

181 line: reference

208 line: add that method described in 2.1 section

210 line: add BBCH

214 line : BBCH

240 line: reference

296 line:  in 2.1 section

324 line: reference

333 line: reference

349,  358, 377, 387, 397, 407, 417, 427, 439 line: how many replicates?

Author Response

Dear reviewer,

Thank you for your constructive suggestions. We followed all your valuable inputs and improved our MS accordingly. The same has been uploaded for your consideration.

The response for some of your major quarries are as follows:-

Q.1. 164 line: add a table of bacteria collected.

Response: Firstly we screened all 13 isolates of endophytic bacteria though the antagonistic activity in dual culture plate assay. Afterward we find single bacterium which possess enormous potential to control the C. truncatum that’s why we proceed a single bacterium for their molecular analysis. Subsequently isolated bacterium was identified as P. dendritiformis.

Q.2. 171 line: bacteria were identified by PCR or not? or only selected for future works?

Response: Yes, bacterium was confirmed by PCR technique. We don't write the detail procedure of molecular analysis in 2.3.3 because of lengthy manuscript. 

Q.3. 174 line: add a table of selected bacteria used in this study. You can make one table together with all 13 selected.

Response: I have only one bacterium i. e. P. dendritiformis

Q.4. You used only one strain of bacteria for future works?

Response: Yes

Q.5. 210, 214 line: add BBCH

Response: BBCH of chilli was mentioned